# Data-driven rolling model for global wave height

Xinxin Wang [3], Jiuke Wang [4], Wenfang Lu [5], Changming Dong [6], Hao Qin [3], Haoyu Jiang *[1,2,3]

[1] College of Life Sciences and Oceanography, Shenzhen University, Shenzhen, China
[2] Laboratory for Regional Oceanography and Numerical Modeling, Qingdao Marine Science and Technology Center, Qingdao, China
[3] Shenzhen Research Institute, China University of Geosciences, Shenzhen, China
[4] School of Artificial Intelligence, Sun Yat-Sen University, Zhuhai, China
[5] School of Marine Sciences, Sun Yat-Sen University, Zhuhai, China
[6] School of Marine Sciences, Nanjing University of Information Science & Technology, Nanjing, China

Corresponding author: Haoyu Jiang (Haoyujiang@szu.edu.cn)

**Abstract.** Significant Wave Height (SWH) is crucial for many human activities, such as marine navigation, offshore operations, and coastal management. Traditionally, SWH is modeled using numerical wave models, which, while accurate, are computationally intensive and constrained by incomplete physical representations of wave spectral evolution. This study introduces a simple global deep learning-based model for SWH, which uses the current SWH field and the wind field at the next time step as inputs to predict the SWH field at the next time step. This approach mirrors the rolling prediction strategy of numerical wave models. After training on a re-analysis dataset, the errors of the model accumulate lightly with time when given a good initial field because no spectral information is used. However, after accumulating for ~200 hours, the errors stabilize, remaining comparable to those of state-of-the-art numerical wave models. Additionally, the error accumulation can be mitigated through the assimilation of altimeter measurements. This deep learning model can not only serve as an efficient surrogate for traditional numerical wave models with respect to SWH but also provide a baseline for statistical modeling of global SWH due to its simplicity in inputs and outputs.

## 1 Introduction

Wind-generated surface gravity waves (hereafter, waves) are one of the most common physical phenomena on the sea surface. These waves impact nearly all human activities in the ocean, including ocean engineering, maritime navigation, fisheries, and port operations. Moreover, ocean waves play a crucial role in many geophysical processes at the sea surface, such as the exchange of mass, momentum, and energy within the wave boundary layer. Thus, it is essential to keep improving our ability to model ocean waves.

Numerical Wave Models (NWMs) are the most widely used tool for forecasting and hindcasting waves. These models apply numerical methods to solve wave action balance equations, thereby representing the evolution of wave spectra. Over years of development, widely used NWMs like WAVEWATCH III (WW3) (Tolman, 1991; Tolman et al., 2002) and SWAN (Simulating Waves Nearshore) (Booij et al., 1999) have demonstrated the capability to provide spatio-temporal distributions

of wave parameters, such as Significant Wave Height (SWH), given reliable wind forcing fields (e.g., Alday et al., 2021; Liu et al., 2021).

However, NWMs have certain limitations. While they have been successfully providing operational wave forecasts for decades, their computational cost can still be a challenge, particularly for high-resolution simulations. The evolution of wave spectra in

NWMs occurs within a five-dimensional space (two spatial dimensions, time, frequency, and direction), which increases the complexity of numerical computations. Additionally, the accuracy of NWMs is constrained by incomplete physical representations and numerical effects.

The rapid development of artificial intelligence (AI) offers potential solutions to the limitations of traditional NWMs. Recent

advancements in AI weather forecasting have demonstrated that AI-based models can achieve better accuracy than numerical models with much lower computational costs (e.g., Lam et al., 2023, Bi et al., 2023), providing the confidence for developing AI-based wave models. Consequently, some studies have already explored AI applications in wave modeling. Some have attempted to replicate the AI weather forecasting approach by treating wave modeling as a purely nonlinear auto-regression problem of spatio-temporal series (e.g., Zhou et al., 2021; Ouyang et al., 2023). However, this approach

overlooks the fact that phase-averaged wave modeling should not be treated as an initial value problem. Without a wind field driving the model, it is physically impossible to accurately simulate waves directly from past wave evolution alone. While initial conditions do play a role in short-term prediction, these auto-regression models cannot even run without the initial conditions provided by an NWM.

Recent studies have adopted a rolling SWH prediction strategy similar to NWMs, utilizing both initial SWH fields (past and present) and forcing wind fields (future winds) as inputs, with future SWHs as outputs. However, most of these studies have focused on wind-sea-dominated nearshore areas (e.g., Cao et al., 2023; Gao et al., 2023), where swell propagation is not a dominant factor in wave modeling. These studies have found that the error in these AI models increases over time compared to NWM hindcasts. This is not surprising because the models do not account for spectral information, and different spectra

with the same SWH respond differently to the same forcing. If such an error accumulation is too large, the AI model will not be able to run independently without the initial SWH field from NWMs. Conversely, if the error accumulation is minor, the model may still be valuable for various applications. However, to the best of our knowledge, no study has yet discussed whether such a model combined wind and SWH inputs can operate effectively using a rolling strategy without relying on NWM data.

One potential solution to solving this problem in AI wave modeling is straightforward, that is, to incorporate the full directional wave spectrum, allowing the AI to approximate the solution of the wave action balance equation. However, applying this method to global SWH modeling presents significant challenges. The global directional wave spectra at any given moment

form a very large 4-D matrix. When using these 4-D matrices as inputs and outputs of an AI model, the training will require an enormous dataset and a complex model architecture.


From an engineering perspective, the accumulation in model simulation errors can be mitigated through data assimilation. In NWMs, the assimilation of altimeter-measured SWH does not always yield positive outcomes because altimeters provide only wave height information without detailed wave spectra (e.g., Ardhuin et al., 2019; Jiang et al., 2022). However, it is worth investigating whether the assimilation of altimeter data can enhance the accuracy of AI-based SWH modeling.


In this study, we propose a global-scale deep learning-based model for SWH. The model utilizes a rolling prediction strategy, similar to NWMs, by taking the current SWH field and the wind field at the next time step as inputs and predicting the SWH field at that next time step. This model is designed to address two key questions: 1) How does a simplified global AI wave model, using an input-output framework similar to NWMs but without incorporating spectral data, handle error accumulation? 2) Can the assimilation of altimeter data help reduce error accumulation and improve the reliability of SWH modeling?


After training the model on a re-analysis dataset, it was observed that, as expected, the AI model experiences a slight accumulation in error over time when provided with a good initial field. However, after approximately 200 hours, the error stabilizes, and the stabilized errors are not significantly larger than those of state-of-the-art NWMs, which is somewhat surprising. Additionally, we demonstrate that the issue of error accumulation can be partially mitigated through the assimilation of altimeter measurements. Although good results have been obtained by the AI model presented in this study, it is noted that we do not intend to suggest that the AI model is superior to traditional NWMs or that it could replace NWMs. NWMs still retain numerous advantages over AI approaches, such as their ability to provide parameters beyond SWH and their stronger physical interpretability, among other merits. The AI model we have developed should be more regarded as a model surrogate specifically for time-sensitive or computation resource-sensitive scenarios. The remainder of this paper is organized as follows: Section 2 describes the data and methodologies employed in this study. Section 3 presents the results from the AI model and their evaluation, followed by discussions and conclusions in Section 4.



## 2 Materials and Methods

### 2.1 Data


#### 2.1.1 ERA5 Wind and Wave Data

The ERA5 is a comprehensive global climate reanalysis dataset, covering the period from 1950 to the present, with hourly data on a wide range of atmospheric and wave parameters (Hersbach et al., 2020). This dataset is based on state-of-the-art modeling technology and has assimilated global historical observations to produce global estimates of these parameters. The

wave data in ERA5 is derived from the Wave Model (WAM) hindcast and has assimilated SWH data from various altimeters,
including ERS-1/2, ENVISAT, JASON-1/2, CRYOSAT-2, and SARAL, using an optimal interpolation scheme. This
assimilation enhances the accuracy of SWH data, particularly in the open ocean, making ERA5 more reliable compared to
other NWM hindcasts. Due to its accuracy and consistency, ERA5 data products have been widely utilized in wave-related
research (e.g., Jiang and Mu, 2019; Jiang, 2020). The dataset is available through the Climate Data Store, with pre-interpolated
resolutions up to $0.25° × 0.25°$ for atmospheric parameters and $0.5° × 0.5°$ for wave parameters.

In this study, we utilized the global SWH and 10-meter longitudinal and latitudinal components of neutral wind ($U_{10}$ and $V_{10}$)
from the ERA5 dataset for the period 2000-2017 to train the global AI SWH model. The corresponding data in the year 2022
was used for validation to alleviate over-fitting, while the model testing was conducted with data in the year 2020. In addition,
we used the swell SWH data also from ERA5 to analyse the impact of swells on the model performance. Both the wind and
wave data used here are at a $0.5° × 0.5° × 1h$ spatio-temporal resolution.

### 2.1.2 CCI-Sea State Dataset

The altimeter dataset used in this study for data assimilation experiment and model evaluation is the Climate Change Initiative
(CCI)-Sea State dataset version 3 (Dodet et al., 2020). This dataset provides accurate and consistent global SWH data. The
SWH data have undergone rigorous quality control and joint calibration to minimize systematic errors across altimeters.
Additionally, a non-parametric empirical mode decomposition technique has been employed for data de-noising, effectively
reducing random measurement errors. As shown by Jiang (2023), after reducing random noise, the typical error of SWH from
the CCI-Sea State is only ~0.15 m in the open ocean, making the dataset well-suited for calibrating and evaluating SWHs from
NWMs. To minimize land contamination, altimeter measurements within 50 km offshore were excluded from the dataset.

### 2.1.3 WAVEWATCH-III Hindcast

The SWH data from the WAVEWATCH-III model hindcast with the physical parameterizations by Liu et al. (2021), hereafter
referred to as WW3-ST6, were utilized as a benchmark to evaluate the performance of the AI model. This hindcast is driven
by ERA5 10-m surface winds and has a spatial-temporal resolution of $0.25° × 0.25° × 3h$. Although it does not assimilate wave
observations, the WW3-ST6 hindcast shows good agreement with observational data, achieving an overall RMSE of
approximately 0.35 m (or 5%–15% of SWH) compared to altimeter data in the open ocean. Detailed information and access
to the dataset can be found in Liu et al. (2021).

### 2.1.4 NDBC Buoy Dataset

To further validate the proposed model, we utilize independent in situ observations from the NDBC (National Data Buoy
Canter) buoy dataset. These buoys provide accurate in situ SWH measurements at specific ocean locations, serving as a reliable
reference for model evaluation. Similar to the selection criteria for altimeter measurements, we mitigate coastal effects by
using several NDBC buoys located at least 100 km offshore.

### 2.2 Deep Learning Model

**2.2.1 Model inputs and outputs**

The deep learning model for SWH in this study employs an input-output structure similar to NWMs. The SWH field at any
time point $T_i$ (initial SWH field) and the wind field ($U_{10}$ and $V_{10}$) at the next time point (one hour later in this case) $T_{i+1}$ are
used to predict the SWH field at $T_{i+1}$. The model can then further predict the SWH field at $T_{i+2}$ using the SWH field at $T_{i+1}$ and
the wind field at $T_{i+2}$, which is a rolling prediction strategy. We understand that adding historical wind information might
enhance the accuracy of the AI SWH model. Particularly, if a long series of wind fields are used as inputs, the model can work
in a different way that the initial SWH field is not needed (e.g., Song and Jiang, 2023; Wang and Jiang, 2024). However, one
of our aims at this stage is to maintain the model's similarity to NWMs to test the effectiveness of this straightforward and
simple input-output structure.

### 2.2.2 Model Structure

This study employs a U-Net architecture for the AI modeling of global-scale SWH. U-Net is a convolutional neural network
(CNN) originally designed for biomedical image segmentation. It is characterized by its U-shaped structure, which combines
an encoder and a decoder through skip connections (Ronneberger et al., 2015). The encoder progressively extracts features
from the input through convolution and pooling, while the decoder reconstructs spatial resolution using de-convolution and
up-sampling. Skip connections link corresponding layers of the encoder and decoder, preserving high-resolution details. Such
a CNN-based deep learning model is well-suited for wave statistical modeling using our input-output structure, and the
effectiveness of U-Net in in wave modelling has been shown in previous studies (e.g., Gao and Jiang, 2023; Wang and
Jiang, 2024).The processes of both local wave generation by wind and wave propagation in space can be captured by
convolutional kernels at different scales.

Figure 1 presents a schematic of the U-Net architecture used in this study. The input matrix consists of three channels: the
global SWH field at $T_i$ and the $U_{10}$ and $V_{10}$ fields at $T_{i+1}$. To handle the wraparound at the -180° and 180° longitude boundary,
we employed an engineering trick of extending the input fields from -180° to 180° (720 longitudes) to -190° to 190°
(760 longitudes). Specifically, the data from -180° to -170° were duplicated and appended to 180° to 190°, and a similar

treatment was applied to the opposite boundary. This effectively connecting the two boundaries and avoiding discontinuities during the modeling process. The final output, the SWH field at $T_{i+1}$, also spans -190° to 190°, but only data from -180° to 180° were retained in the computation of the cost function for training.

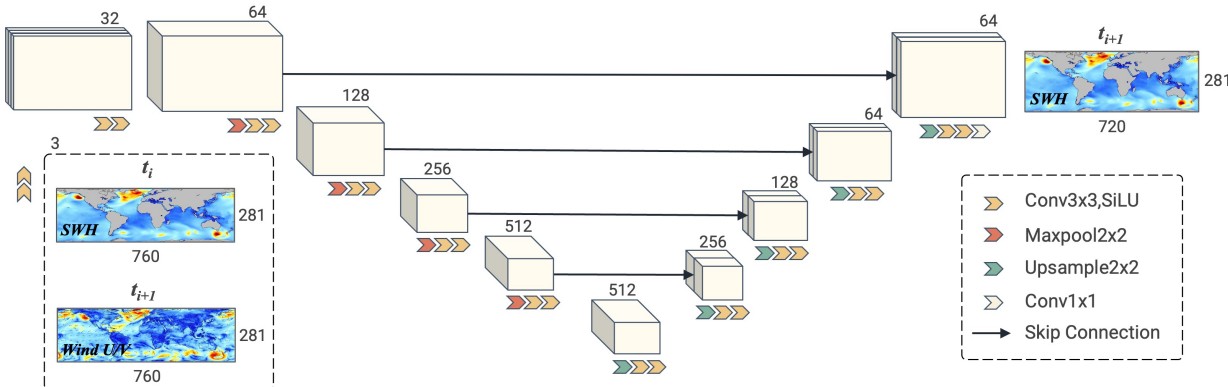

**Figure 1:** An illustration of the U-Net architecture used in this study. Each cube represents a feature map, with the numbers on the sides indicating the number of channels. The legend on the lower-right panel explains the meaning of the different arrows used in the schematic diagram.

### 2.2.3 Model Training

For model training, the training set was randomly shuffled, and the model was then trained to minimize the Mean Squared Error (MSE) between the model output and the target output:

$$Loss = \frac{1}{n}\sum_{j=1}^{Lat}\sum_{i=1}^{Lon}\left[\left(y_{i,j} - x_{i,j}\right)\cos\theta_j\right]^2 \qquad (1)$$

where $x$ and $y$ denote the SWH from the AI SWH model and ERA5, respectively; the subscripts $i$ and $j$ denote the $i$-th longitudinal and $j$-th latitudinal grid point; $\theta_j$ denotes the latitude of the $j$-th latitudinal grid point. This cosine term was introduced to account for the area change of grid points with latitudes. We used six batches for training and trained the model for up to 30 epochs at a learning rate of 0.0001 using the AdamW optimizer. To alleviate overfitting, we implemented a commonly used deep learning technique where training is halted when the loss in the validation set does not decrease for four epochs. Using our training samples (data from 2000 to 2017), training took approximately one hour per epoch on an NVIDIA RTX 4090 GPU. It is also tested that the model's performance will be slightly worse if the training samples are significantly reduced. Once trained, the model requires less than 10 minutes to compute (infer) the global SWH for one year at a spatio-temporal resolution of 0.5° × 0.5° × 1h on an NVIDIA RTX 3060 GPU.

### 2.2.4 Epoch Ensemble Method

To further improve the accuracy and stability of the model predictions, this study employs the epoch ensemble method. This approach mitigates potential issues like over-fitting or under-fitting, which can arise from relying on a single model, by leveraging the diversity of models trained across different epochs. The simplest way of using this method is to retain several models obtained in different epochs during the training process and average their outputs during inference. This straightforward

yet effective strategy enhances model performance without requiring additional training. In this study, the ensemble size was set to four, and the ensemble mean reduced the Root-MSE (RMSE) by ~30% compared to individual models. The final AI models established in this study are available from the Zenodo repository at: https://zenodo.org/records/14244062 (Wang, 2024).

### 2.2 Error Metrics

The Bias, RMSE, Correlation Coefficient (CC), and Scatter Index (SI) are used as the error metrics to evaluate the performance of the AI SWH model, which are defined as:

$$Bias = \frac{1}{n}\sum_{i=1}^{n}\left(y_i - x_i\right) \tag{2}$$

$$RMSE = \sqrt{\frac{1}{n}\sum_{i=1}^{n}\left(y_i - x_i\right)^2} \tag{3}$$

$$CC = \sum_{i=1}^{n}\left(y_i - \bar{y}\right)\left(x_i - \bar{x}\right)/\left[\sqrt{\sum_{i=1}^{n}\left(y_i - \bar{y}\right)^2}\sqrt{\sum_{i=1}^{n}\left(x_i - \bar{x}\right)^2}\right] \tag{4}$$

$$SI = RMSE / \bar{y} \tag{5}$$

where $x$ and $y$ denote the SWH from the AI models and reference data (which can be either ERA5 or CCI-sea state altimeter

data), respectively; $n$ is the sample size, and the bars over $x$ and $y$ denote their mean values. These error metrics were also used to monitor the training process of the model.

### 2.3 Data Assimilation

To reduce error accumulation in the long-term operation of the rolling prediction, we tried to incorporate data assimilation techniques by integrating altimeter measurements to correct the model's "initial" SWH field. It is noted that in our input-output

setting, the outputs of the last time step will be the "initial" SWH field of the next time step. In the assimilation of NWMs, spectral information is used so that it requires a method to transform observations of SWH or other integrated wave parameters into wave spectra. However, in our case, only SWH fields and wind fields are used as the model inputs, without involving spectral information. Thus, the assimilation of this model also does not need to involve the spectral information, which simplifies the assimilation.

We employed a simple objective analysis (OA) method, which is a form of optimum interpolation (OI), for data assimilation:

$$A(i,j,t) = M(i,j,t) + \sum_{k=1}^{N} w(i,j,t)_k \, g(O_k - M_k) \tag{6}$$

$$w(i,j,t)_k = \exp\left[-d_k^2(i,j,t)/2R(i,j,t)^2\right] \Big/ \sum_{k=1}^{N} \exp\left[-d_k^2(i,j,t)/2R(i,j,t)^2\right] \tag{7}$$

$$R(i,j,t) = \min\left[d_k(i,j,t)\right] \quad (k=1,2,3,...,N) \tag{8}$$

$$d_k(i,j,t) = \sqrt{(S_k/S_1)^2 + (T_k/T_1)^2} \quad (S_k < 1500\,km, T_k < 48\,h) \tag{9}$$

where $i$, $j$, and $t$ represent longitude, latitude, and time, respectively; $M$ (model) and $A$ (assimilated) represent the model outputs before and after assimilation, respectively; $k \in \{1,2,...,N\}$ represents the number of observations to be assimilated at a given time; $O_k$ and $M_k$ represent the values of observed and corresponding modeled SWH at the spatio-temporal location of the $k$-th observation; $w_k$ represents the weight factor for correction at location $(i, j, t)$ for the $k$-th observation; $d_k$ represent the spatio-temporal distance from location $(i, j, t)$ to the $k$-th observation; $R(i,j,t)$ represent the distance from location $(i, j, t)$ to its nearest observation. $S_k$ and $T_k$ are the spatial and temporal differences between the location $(i, j, t)$ and the $k$-th observation, respectively, and $S_1$ and $T_1$ are tuning coefficients to combine spatial and temporal distances. Previous studies often used a 30-min-50-km window to collocate SWH from altimeters and other sources (e.g., Jiang, 2020), thus, $S_1$ and $T_1$ are set to 50 km and 0.5 h, respectively.

Here the simple OA method is used instead of the more complex variational methods primarily because OA is significantly less computationally demanding than variational methods. One of the advantages of the AI model is its efficiency and lightweight nature. Introducing variational methods for assimilation would increase computational demands by several orders of magnitude, rendering the AI model inefficient and impractical. Furthermore, OI enables incremental assimilation of observations, allowing for continuous updates as new data becomes available. In contrast, variational methods typically require a complete assimilation cycle, which may not be feasible for fast-paced AI applications. Besides, it is noted that there is no clear evidence that variational methods outperform OI in wave modeling.

In our data assimilation experiment, assimilation was conducted every six hours (i.e., every six time steps, observations are used to corrected the outputs of the rolling model and the updated outputs are used as the new inputs at the next time step), beginning after the first 24 hours of the model run. Of course, the frequency of data assimilation can be user-defined. A higher assimilation frequency generally leads to more accurate results but also entails increased computational costs, and vice versa. During each assimilation, the SWH data from the CCI-Sea State dataset were used to correct the AI model's hindcasts using Equations 6-9. It is noted that in Equation 9, the upper limits of $S_k$ and $T_k$ mean that only observations within 1500 km can influence the value of the target grid point. Only observations from the past 48 hours were used to correct the current SWH field. After assimilation, the prediction for the next time step used the assimilated SWH field as inputs for the AI model.

# 3 Results

## 3.1 Temporal Stabilization of Model

The performance of the proposed rolling AI model for SWH was evaluated on the 2020 test dataset. We selected initial SWH fields every 36 hours from 00:00:00 2020-Jan-1 (i.e., the 0[th], 36[th], 72[nd],…, 8460[th] hours of 2020, totalling 236 sets of experiments). For each initial SWH field, a 300-hour rolling modeling was conducted. Figure 2 shows the variation of global overall error metrics compared to ERA5 SWH with simulation time. The orange and blue lines represent the mean values of the error metrics for the 236 experiments before and after assimilation, respectively, with the shaded areas indicating the range of these metrics across different starting times. Note that evaluating against other untrained years yields similar results, with differences in correlation coefficient (CC) and root mean square error (RMSE) being less than 0.003 and 0.03, respectively.

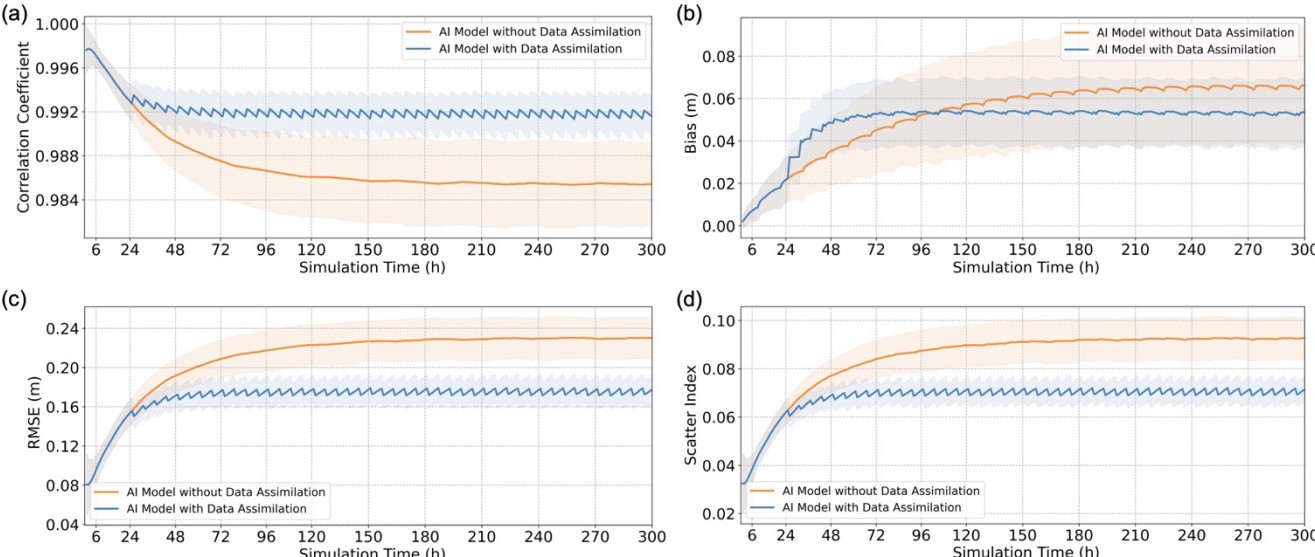

**Figure 2:** The variation of global overall error metrics between the AI SWH model outputs and ERA5 with simulation time: (a) CC, (b) bias, (c) RMSE, and (d) SI. The orange and blue lines represent the mean values of the error metrics for the 236 experiments starting from different initial SWH fields, before and after assimilation, respectively. The shaded areas around the lines indicate the range of error metrics across different experiments with varying initial SWH fields.

For the condition without assimilation, the curves for all four error metrics show that the errors of the AI SWH model increase rapidly within the initial 48 hours of simulation time. As mentioned in the introduction, this trend is expected given the absence of spectral information. However, as simulation time progresses, the rate of error growth diminishes, and the model stabilizes after ~240 hours. This means the model can still capture some aspects of SWH evolution over time. Remarkably, the global overall mean values for CC, bias, RMSE, and SI are around 0.985, 0.06 m, 0.23 m, and 0.09, respectively, comparable to state-of-the-art NWMs. This suggests that the simple AI model can work without the assimilation of observation and the

255 information from NWMs, at least, in some applications such as modelling the SWH in wind-sea dominated regions (see section 3.2.2).

When data assimilation is applied, the errors are significantly reduced across all metrics, except for bias before the ~100th hour. The increase in bias is likely due to minor inconsistencies between ERA5 and CCI-Sea State, and the bias remains less 260 than 0.06 m. For the other error metrics, assimilation reduces the time required for error stabilization to ~72 hours while also lowering the final converged errors of the AI model. When stabilized, the global overall CC, bias, RMSE, and SI reach 0.992, 0.05 m, 0.17 m, and 0.07, respectively. Although these metrics are calculated relative to ERA5 data rather than direct observations, these values seem to be completely acceptable for most operational wave modeling applications.

265 To further test whether such a model can operate independently, we conducted a "cold start" experiment using an initial SWH field of zero. The results, shown in Fig. 3, are compared with the "hot start" results from Fig. 2. Although the initial SWH fields are the same (zeros) in the "cold start" experiment, varying wind fields at different starting times lead to differences in error metrics (depicted by blue shadows in Fig. 3). As expected, the cold start experiment shows larger errors initially, but these errors diminish over time, converging to values similar to those from the "hot start" after approximately 240 hours. This 270 convergence demonstrates the robustness of the model.

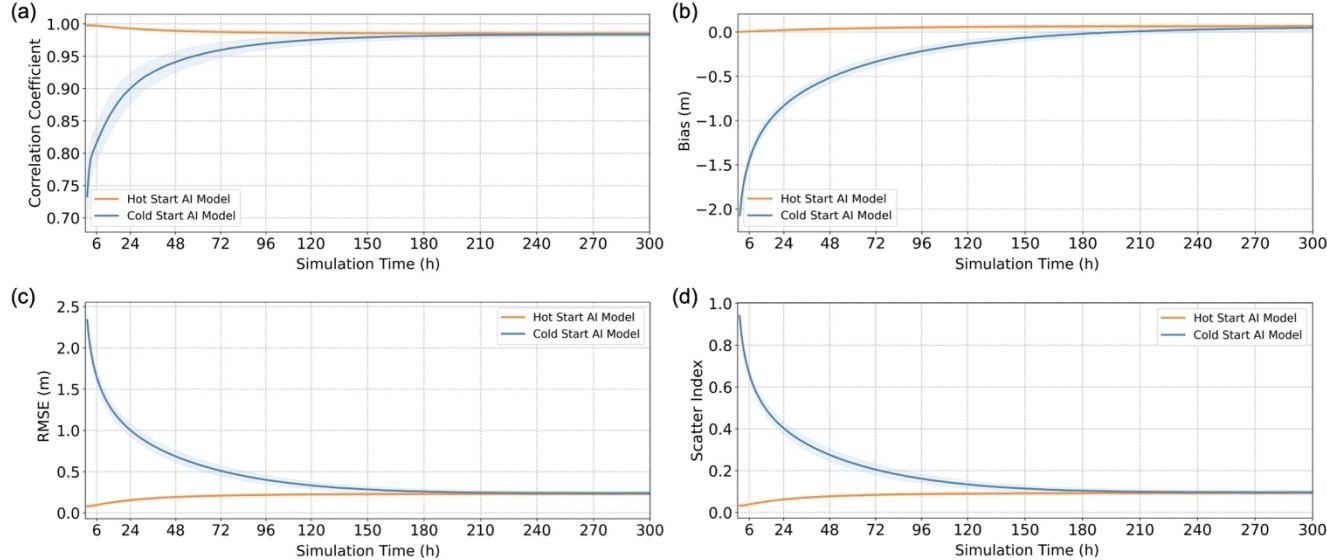

**Figure 3:** The variation of global overall error metrics between the AI SWH model outputs and ERA5 with simulation time: (a) CC, (b) bias, (c) RMSE, and (d) SI. The orange lines and shaded areas are the same as those in Fig. 2. The blue lines and shaded area are the corresponding results for the cold start with an initial field of zero SWH. These results do not use data 275 assimilation.

## 3.2 Spatial Distributions of Model Errors

To further identify and understand conditions where the AI model performs well, we compare the SWH produced by the AI model against different sources of SWH and show the geographical distributions of the four aforementioned error metrics for both results with and without data assimilation. The results for 6-h, 24-h, 72-h, and 240-h simulations without data assimilation are shown in Fig. S1, S2, and S3 of the Supporting Information (SI) and Fig. 4, respectively. Although errors increase with simulation time, as shown in Fig. 2, these results indicate that the spatial error patterns remain consistent across different simulation time. Our primary focus is on the 240-h hindcast results in Fig. 4, where the errors have stabilized. Also, since high-quality SWH initial field like ERA5 is not always easily available, the stabilized error is more typical and meaningful as the reference when the AI SWH model is run independently.

### 3.2.1 Evaluated against ERA5

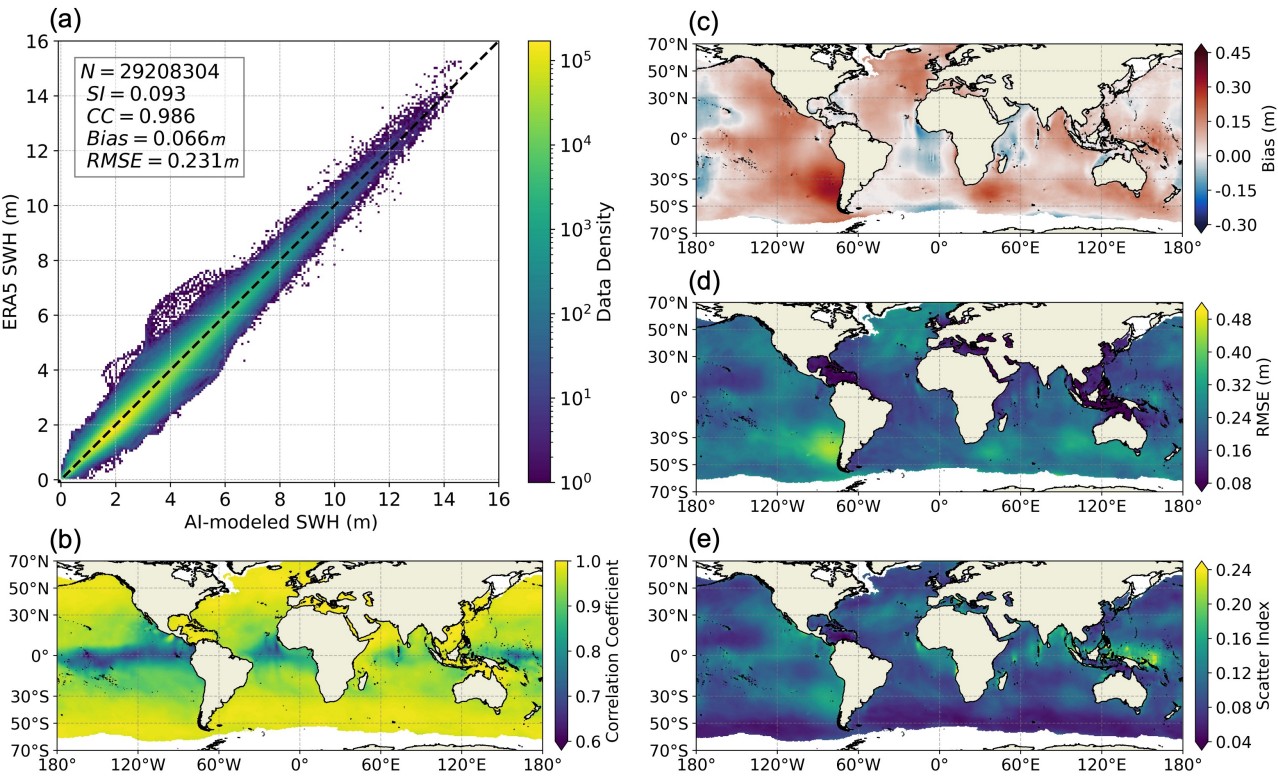

**Figure 4:** Comparison of SWHs from the AI model (without data assimilation) at 240-h hindcast time (when the errors are stable) with ERA5 for the year 2020. (a) Scatter plot between the SWHs from the two datasets. (b-e) Global spatial distributions of CC, bias, RMSE, and SI, respectively.

The scatter plot in Fig. 4a shows a good overall agreement between the SWHs generated by the AI model and those from ERA5, with most points closely aligning with the 1:1 line. The SI of 0.093, the CC of 0.986, and the RMSE of 0.23 m are already better than those typically observed between contemporary global NWM hindcasts and altimeter data. Although such a direct comparison might not be entirely fair or reasonable, these values indicate that this simple AI SWH model is capable of effectively modeling the distribution and variability of global SWHs.

Regarding the spatial distributions of errors, the CCs (Fig. 4b) are close to 0.99 in the westerlies of both hemispheres and in marginal and (semi-)enclosed seas where wind-seas occur frequently dominated. However, in the tropical oceans, especially along their eastern coasts where swells are predominant ("swell pools", Chen et al., 2002), the CCs are below 0.9 (~0.85 in the Indian Ocean, ~0.8 in the Atlantic Ocean, and ~0.7 in the Pacific Ocean). To further examine whether these results are related to the presence of swell, we examined the relationship between the swell energy proportion (i.e., the ratio of the square of the swell SWH to the square of the total SWH) and the CC across the global ocean (the orange line in Fig. 5). The results show a clear trend: the smaller the swell energy proportion, the higher the CC. In particular, when the proportion is below 0.7, the CC values are consistently above 0.99, indicating robust model performance in wind-sea-dominated regions. However, when the swell energy proportion exceeds 0.7, the CC values for the model without data assimilation drop significantly, corroborating its lower performance in swell-dominated regions.

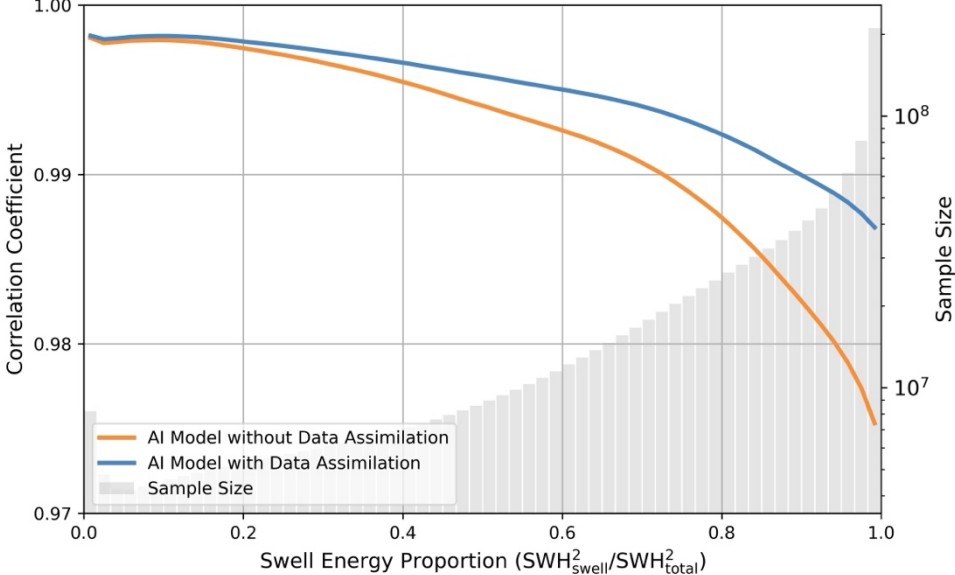

**Figure 5:** Correlation coefficient between AI model and ERA5 data as a function of swell energy proportion the global ocean in the year of 2020. The orange and blue lines represent the AI model before and after data assimilation, respectively, and the grey bars indicate the variation in sample size as a function of swell energy proportion.

Three main factors contribute to these lower CC values in the swell pools. First, the wind-sea growing process can be regarded as a forcing problem while swell propagation is more of an initial value problem. This difference is evident in the CCs observed over different simulation time. For example, in regions of westerlies, the CCs remain stable at around 0.99 across 6-h, 24-h, 72-h, and 240-h hindcasts (Fig. S1b-S3b and Fig. 4b). Conversely, in the Pacific swell pool, CCs decrease significantly with simulation time: from 0.98 at 6 hours, to 0.92 at 24 hours, to 0.8 at 72 hours, and finally to 0.7 at 240 hours. Compared to the wind-sea growth, it is far more challenging, if not physically impossible, for the AI model to accurately learn the swell propagation process using only the evolution of SWH spatial patterns without directional wave spectra. Despite this limitation, the AI model still manages to capture some rough characteristics of swell energy propagation from the SWH data, which is why its performance is still reasonable in these swell-dominated regions.

Second, SWHs in swell pools typically vary within a narrow range of approximately 0.5~3.5 m. This limited variability means that even if absolute RMSEs are relatively low, the CC values may still be low, making it challenging to achieve high CC values. These regions also exhibit the lowest CCs in the comparisons between other global SWH data, i.e., NWM hindcast versus altimeter observations. For reference, comparisons between SWHs from WW3-ST6 and CCI-Sea State are shown in Fig. S4, where CCs in these swell pools are also lower than 0.8.

Third, the Garden Sprinkle Effect (GSE), a numerical error associated with swell propagation, can introduce "random" errors into SWHs when swells have propagated over large distances. Such swells are very common in swell pools and it is probably impossible for the AI model to learn how these numerical errors evolve using the ERA5 SWH data.

Regarding the bias, the values vary in the range of ±0.15 m in most parts of the ocean but can reach 0.3 m to the Southwest of South America and 0.2 m to the Southeast of Africa. These relatively large biases are related to the accumulation of error with simulation time. It is not clearly known why the bias has such a distribution. We also plotted the distribution of bias in other years and found that the regions with the largest bias are slightly different in different years but the overall patterns are similar. For example, the results in the year 2000 are shown in Fig. S5 where the error maps look similar to those in Fig. 4. It is noted that although the data from the year 2000 is used in the model training, the training is only based on 1-h simulations without rolling so the AI model has never "seen" the exact input for the simulation time of more than two hours. It is not a wrong way to use the data in the training set to do a long-term rolling test (of course, using an independent testing set should be better). In all years, these biases are not significant compared to the typical annual mean SWH in the corresponding regions. Besides, these biases can largely be corrected by some simple post-process methods such as point-by-point linear regression.

The RMSE pattern in Fig. 4d shares some similarities with the bias pattern, with the largest RMSE values also found southwest of South America, indicating that bias significantly contributes to the overall error. However, the swell pools with relatively low CCs are not prominently visible in the maps of bias, RMSE, and SI. In terms of SI, apart from the regions with relatively

large RMSE, high SI values are also observed near small islands and archipelagos such as Indonesia. The annual mean SWHs in these areas are lower than in the open ocean, so even a small RMSE can lead to a relatively large SI. Moreover, these areas represent only a small portion of the global ocean, so their contributions to the overall loss function are minimal. However, wave behavior in these regions differs significantly from that in the open ocean, leading to a different input-output relationship for the model, which complicates the training process. Additionally, NWMs also encounter numerical errors when handling these small islands. These factors make it more challenging for the AI model to effectively "learn" from data near small islands.

The AI model does not directly incorporate ice information, treating ice-covered regions simply as land. As a result, higher errors are expected in polar regions. However, this is not evident in Fig. 4. In contrast, Fig. S1 shows that errors in polar regions increase rapidly at first but then stabilize with simulation time. This pattern may be due to the variability of sea ice, which leads the model to primarily learn the rapid response of SWHs to wind forcing in polar regions. Consequently, SWHs in polar regions are less sensitive to the initial field compared to other areas. It is also noted that only the data outside the marginal ice zone is used, meaning waves that propagate into these regions are not considered. For waves generated in the marginal ice zone and propagating out, they contribute minimally to the overall SWH energy so they are neglected by the AI model.

The distributions of these error metrics suggest that the AI model performs well across global oceans in general. To provide a more intuitive understanding of the AI SWH model's performance, an animation comparing the global SWH distributions from ERA5 and our AI model is presented in Movie S1 in the Supporting Information. Slightly different from the 240-h hindcast results in Fig. 4, the results in Movie S1 are generated by continuously rolling the AI model from 01-Jan-2020 00:00:00. A simple visual inspection of the movie indicates that the AI model effectively captures SWH evolution, suggesting that the AI model could serve as an effective surrogate for NWMs, at least for some wind-sea-dominated regions. Moreover, we also verified that the spatial distribution of error metrics varies with season, which reflects seasonal differences in wave climate. Such variability is consistently observed across statistical models, AI-based models, and traditional numerical wave models (NWMs).

### 3.2.2 Evaluated against altimeter and buoy data

In addition, we compared the SWHs from the 240-h~272-h hindcasts of the AI model with those from the CCI-Sea State to evaluate the model performance, with the results shown in Fig. 6. This direct comparison with altimeter-measured SWHs provides an independent and commonly used method for wave model evaluation. To ensure sufficient collocation between the altimeter and model data, we extended the simulation period by 36 hours, making sure that every altimeter data record in the open ocean can be collocated with a "model grid point".

The comparison shows that the AI model also aligns well with the altimeter data. In Fig. 6a, most data points lie along the 1:1 line, with a bias close to zero, an RMSE of 0.336 m, a CC of 0.968, and an SI of 0.123. These overall error metrics are

comparable to those observed between WW3-ST6 and CCI-Sea State in Fig. S4 where the bias, RMSE, CC, and SI are 0.032 m, 0.326 m, 0.972, and 0.119, respectively.

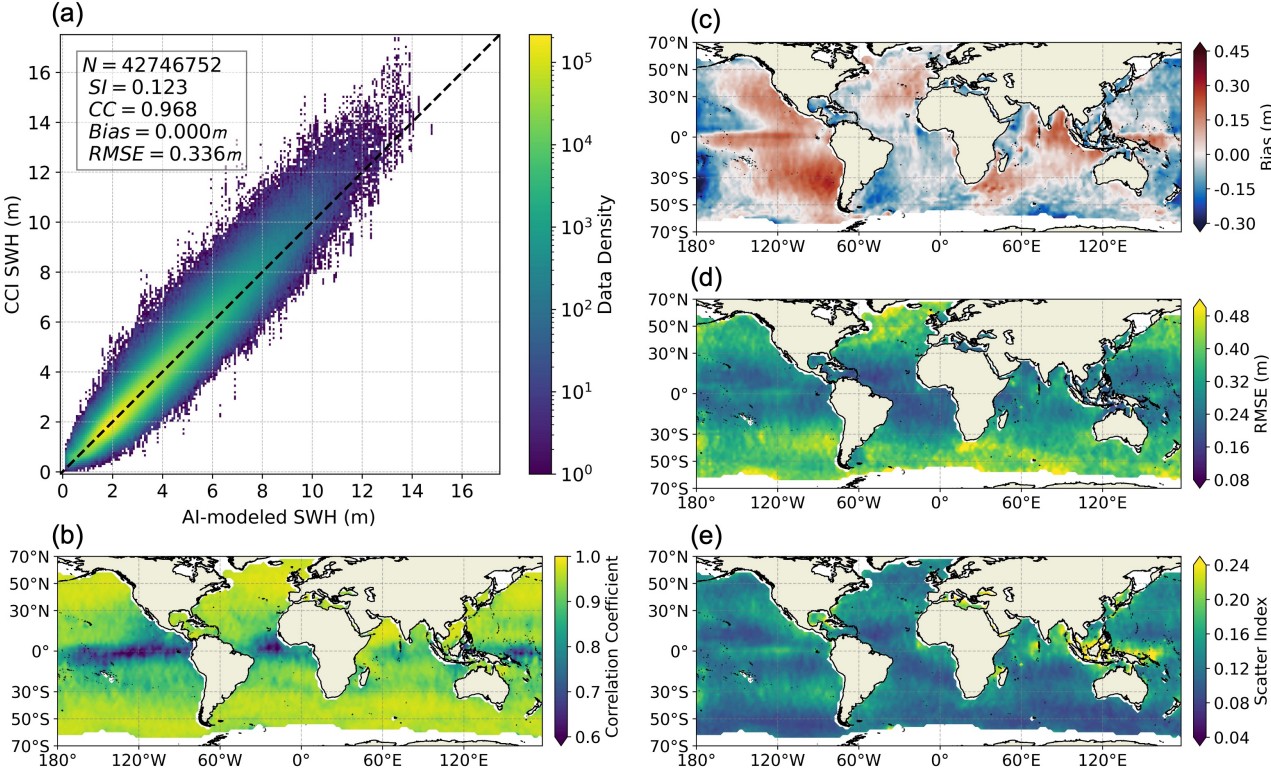

**Figure 6:** The same as Fig. 4, but the comparison is between the 240-h SWH hindcasts of the AI model and the CCI-Sea State dataset.

Regarding the spatial patterns of errors, Fig. 6b-6e are similar to Fig. 4b-4e though the magnitudes of errors are generally larger in Fig. 6. The CCs in Fig. 6b are ~0.98 in the westerlies but are only ~0.6 in the Pacific swell pool. In contrast, the two corresponding CCs are ~0.97 and ~0.7, respectively, in Fig. S4. For other error metrics, the differences between the two models are even smaller. The biases vary in a similar range, and the RMSEs and SIs show similar patterns in Fig. 6 and Fig. S4. Notably, in the westerlies the RMSE and SI values from the AI model are even slightly lower than those from WW3-ST6, a state-of-the-art NWM hindcast. These findings further demonstrate the strong performance of the AI SWH model, particularly in open ocean regions that are not always predominated by swells.

For a more independent and objective assessment of the AI model, we further compared its results with independent in situ observations from NDBC buoys (Fig. 7). Similar to the comparison with altimeter data, most points align well along the 1:1 line, with bias, RMSE, CC, and SI values of -0.002 m, 0.306 m, 0.959, and 0.161, respectively. For reference, Fig. S6 presents

the comparison between WW3-ST6 and NDBC data, where the corresponding bias, RMSE, CC, and SI values are -0.004 m, 0.291 m, 0.963, and 0.153. These results are largely consistent with the comparison against CCI-Sea State data, further the reliability of AI model.

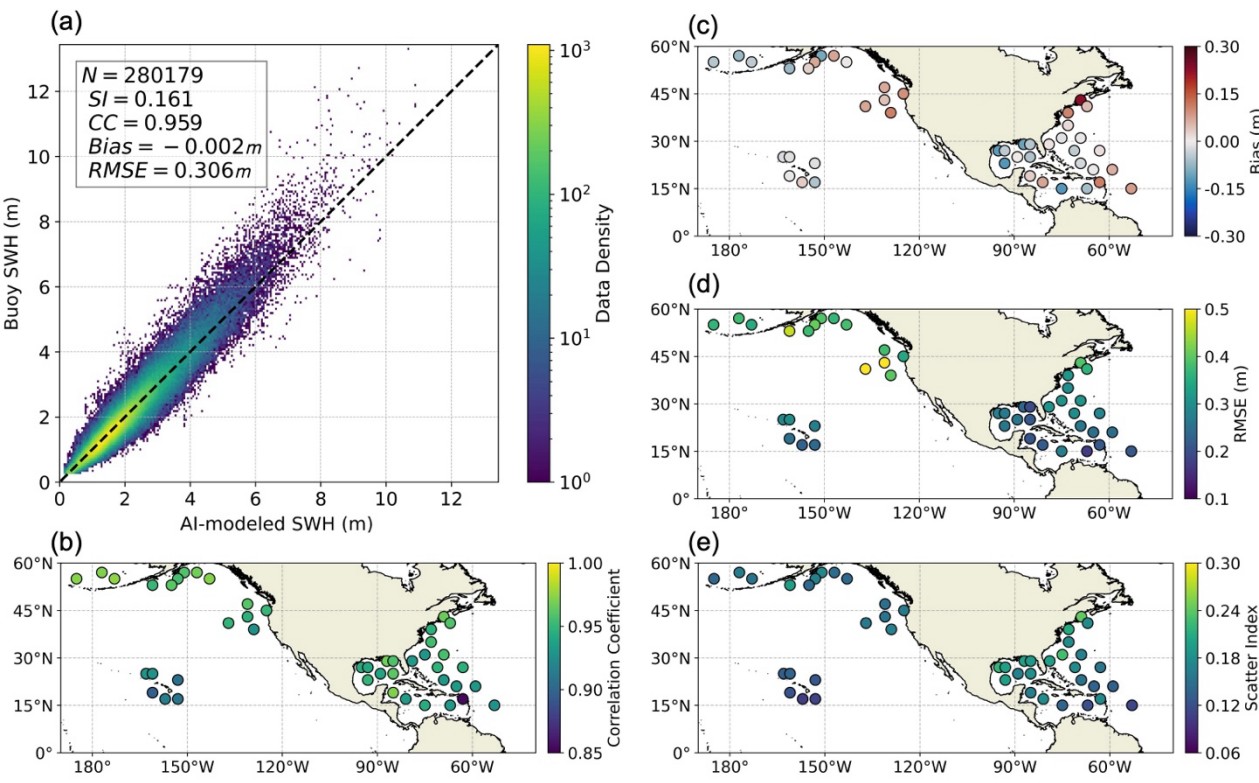

**Figure 7:** The same as Fig. 4, but the comparison is between the 240-h SWH hindcasts of the AI model and NDBC Buoy dataset.

For the 240-h rolling hindcast results of the AI model with data assimilation every six hours, the corresponding comparisons with ERA5 and CCI-Sea State are shown in Figs. S7 and S8, respectively. Compared to the results without assimilation in Fig. 4, all the error metrics of the model improve significantly with data assimilation in Fig. S7. Specifically, the CCs in the Pacific, Atlantic, and Indian Ocean swell pools increase from ~0.7, ~0.8, and ~0.85 in Fig. 4 to ~0.88, ~0.92, and 0.95 in Fig. 7, respectively. The magnitudes of bias, RMSE, and SI also decrease across the oceans with assimilation, although the bias and RMSE remain relatively high in regions to the southwest of South America and southeast of Africa. The comparison between Fig. 6 and Fig. S8 shows a similar result: the overall errors become significantly smaller, particularly in the swell-dominated regions, with assimilation. These results are further supported by Fig. 5, where the model with data assimilation consistently maintains substantially higher CC values, even when the swell energy proportion exceeds 0.7. Similar to Supplementary Movie

S1, the comparison animation of the results with assimilation is placed in Supplementary Movie S2, where the AI model better captured the SWH evolution. The comparison between NDBC buoy data and the AI model with data assimilation is shown in

Fig. S9, where improvements in all error metrics (bias = 0.033 m, RMSE = 0.279 m, SI = 0.147, CC = 0.967) compared to those in Fig. 7 can also be observed, showing the effectiveness of the assimilation. For reference, comparing ERA5 against NDBC buoy data shows that ERA5 still performs slightly better than our AI model (Fig. S10, with bias = 0.027 m, RMSE = 0.266 m, SI = 0.140, CC = 0.971).

## 4 Discussion

The results demonstrate that it is feasible to develop a usable AI SWH model using only an initial SWH field and the wind field at the next time step as inputs. These are likely the minimum requirements for the inputs of an AI SWH model. As noted in the introduction, relying solely on SWH fields as inputs is insufficient since wind-seas cannot be accurately modeled without wind information. Similarly, one can expect that if the initial SWH field is excluded from the inputs, the AI model would struggle to simulate ocean swells using only the input of the current wind field. To confirm this, we trained an AI model using

only the wind field as input, with SWH at the same time step as the output, with the results shown in Fig. 8. The model performance in Fig. 8 is significantly worse than that in Fig. 4, in both wind-sea- and swell-dominated regions. In areas with frequent wind-seas, such as the westerlies, although the CCs exceed 0.95, the RMSEs can also surpass 0.4 m in both hemispheres, much higher than the values in Fig. 4. In the Pacific and Atlantic swell pools, the CCs are even lower than 0.4. These results underscore the critical importance of including both of the initial SWH field and the forcing wind field as inputs

for the AI SWH model.

From a physical perspective, the SWH at a given location is influenced by the wind speed, fetch, and duration in wind-sea conditions. The wind input provides information on both wind speed and fetch. Meanwhile, duration (or historical wind) information is partially and implicitly conveyed by the SWH input, as it is computed in a rolling simulation using a recursive

method that incorporates past wind data. Although the implicit information provided by global SWHs is not as comprehensive as that from global directional wave spectra, the spatial distribution of SWHs still contains significant historical wind information. This explains why including the SWH input is beneficial for modeling wind-sea-dominated regions and why the AI model can slightly outperform the NWM in these areas.

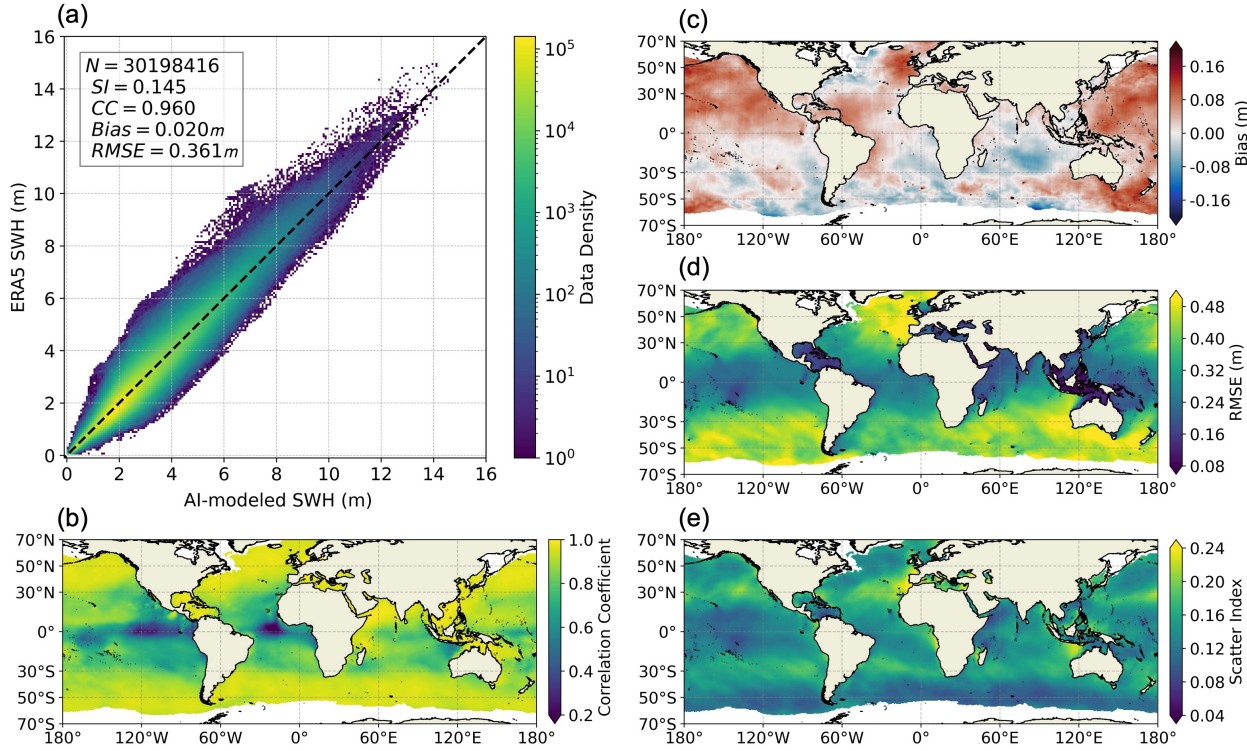


**Figure 8:** The same as Fig. 4, but the AI model is trained only using the wind field at the corresponding time as the input.

In swell-dominated regions, where local wind speeds remain low almost all years, using only the wind input fails to provide any meaningful information about the SWH, as illustrated in Fig. 8. However, as previously mentioned, the AI model can still

learn some rough statistical characteristics of swell energy propagation from the data, especially in regions like swell pools. This is also demonstrated in Movie S1, where the propagation of swells generated by extra-tropical storms into tropical regions is distinctly observable.

Although the above analysis underscores the importance of including SWH input for the AI model, the quality of the initial

SWH is not important if the model is run in a rolling way for relatively long time. The "cold start" experiment has demonstrated that the model error can stabilize within 240 hours, even without an initial SWH field. However, we do not recommend using such a "cold start" in practice because a better initial field or data assimilation can greatly accelerate the speed of error convergence and such a better initial field is almost always available (e.g., using the output of the model in Fig. 8).

Regarding data assimilation, the assimilation of altimeter SWH measurements is sometimes believed not to be always helpful in NWMs, and may even have negative effects in some cases. This is because there are different approaches to using SWH data to correct directional wave spectra, and improper corrections can adversely affect the model results. However, in this AI

model, the spectral information is encapsulated within the SWH, and both the computation and assimilation are directly based on the SWH. Consequently, if the assimilated SWH data is more accurate than the output of the AI model, the assimilation will positively impact the results.

It is not surprising that data assimilation can significantly improve the performance of the AI model, but it is noted that the computational cost of assimilation in this AI model is low. In the assimilation of NWMs, SWH observations are used to correct the spectral densities of directional wave spectra, a four-dimensional array (latitude, longitude, frequency, direction) at a given time step, using empirical relations. In contrast, the assimilation process in the AI model bypasses the need for wave spectral information, requiring corrections only to a two-dimensional SWH array at a given time, also significantly reducing the complexity of the model.

## 5 Concluding Remarks

In this study, a global-scale AI model for SWH is proposed. The model takes the current SWH field and the wind field at the next time step as inputs, and produces the SWH field at the subsequent time step. Such a rolling computation method is similar to that used in NWMs, but the spectral information is not used. As expected, the lack of spectral data leads to an increase in model error during the early stages of the rolling simulation when given a good-quality initial SWH field. However, the rate of error growth slows as the simulation progresses, nearly halting after ~200 hours. More surprisingly, once the error stabilizes, its overall magnitude is not significantly larger than that of state-of-the-art NWMs, particularly under wind-sea-dominated conditions. Although the performance of the AI model in swell-dominated regions is somewhat inferior to that of NWMs, it still produces meaningful outputs, with a correlation coefficient (CC) exceeding 0.7. This suggests that a simple AI model, using only the current SWH field and the wind field at the next time step as inputs, can be practical for many applications, including operational forecasting, at least in regions outside of swell pools.

Additionally, this study demonstrates that the issue of error accumulation can be partially mitigated through the assimilation of altimeter measurements. By using a simple objective analysis method, the assimilation helps the error of the model to stabilize more rapidly and reduces the magnitude of the stabilized error, resulting in a more reliable AI SWH model.

An important advantage of the AI SWH model proposed here is its low computational cost compared to traditional NWMs. For example, on a personal laptop equipped with a single RTX 3060 GPU, the AI model can perform a 1-year global SWH rolling simulation at a resolution of $0.5° \times 0.5° \times 1h$ in just 10 minutes. In contrast, traditional NWMs, such as the WAVEWATCH III model, typically require several days to complete a simulation with the same output, even on supercomputing facilities. This makes the AI model particularly valuable in time-sensitive and resource-constrained scenarios, where it can be used as a surrogate for the NWMs. One potential application of this model is ensemble modeling, both in

operational wave forecasting and wave climate studies. In these applications, it is challenging to run NWMs multiple times using wind fields from different ensemble members of weather forecast models (for wave forecasting) or of various climate scenarios for long-term projection (for wave climate projection) due to the limitation of computational resources. In contrast, these tasks can be efficiently completed using the AI model, even on a standard laptop.

At this stage, the AI model is trained only on SWH data, limiting its applicability to other wave parameters, such as mean wave periods. Developing an AI model for additional wave parameters would require training from scratch with the relevant data. Whether the current model framework, using the corresponding wave parameter at the current time step and the wind field at the next time step as inputs, can be extended to these parameters remains to be tested, which can be one future direction. While we acknowledge the potential for a more refined deep learning architecture to marginally improve model performance, we believe the bottleneck of the current AI model lies in the physics of the input-output relationship. Therefore, it is difficult to further improve the model performance without changing the model inputs.

We have demonstrated that the current SWH field and the wind field at the next time step are minimum requirements for the inputs of an AI SWH model. Such simplicity of model inputs and outputs makes this model a potential baseline for AI-based modeling of global SWH. A promising future direction of this work involves incorporating additional inputs, such as ocean currents and sea ice, into the model. The ultimate goal of this approach, as mentioned in the introduction, would involve using global directional wave spectra at the current time step and the wind field at the next time step as inputs, with the global directional wave spectra at the next time step as the output. Training such a model would be challenging due to the complexity of the task, but ongoing advancements in AI methodologies, particularly in deep learning, are continuously improving the possibility of achieving this goal.

**Author contributions**

Conceptualization: HJ, JW, WL, CD, HQ
Methodology: XW, HJ
Investigation: XW
Visualization: XW
Supervision: HJ
Writing—original draft: HJ, XW
Writing—review & editing: HJ, XW, JW, WL, CD, HQ

**Competing interests**

All authors declare that they have no competing interests.

## Acknowledgments

This work was jointly supported by the National Key Research and Development Program of China (2023YFC3008203), the National Natural Science Foundation of China (42376172), and the Guangdong Basic and Applied Basic Research Foundation (2022A1515240069, 2024A1515012032, 2023A1515240047). We sincerely thank all the reviewers for their valuable comments and constructive suggestions, which have significantly improved this manuscript.

**Code and data availability**

The ERA5 data is downloaded from Copernicus Climate Data (https://cds.climate.copernicus.eu/). The CCI-Sea State dataset is downloaded from the Centre for Environmental Data Analysis (https://archive.ceda.ac.uk/). The NDBC buoy data is available from the NDBC website (https://ndbc.noaa.gov/). The WW3-ST6 dataset is available from Liu et al. (2021), and the subset used in this study is available in the Zenodo repository (https://zenodo.org/records/14244062). The AI models
established in this study and relevant test data have also been archived in the Zenodo repository (https://zenodo.org/records/14244062) (Wang, 2024).

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
