# Peer review of "Data-driven rolling model for global wave height"

_Geoscientific Model Development, 2024_

## Author Comment (AC1)

**Response to Reviewer 1:**

Dear Reviewer:

We would like to thank you for dedicating time to carefully read our manuscript and provide feedback. We sincerely think their detailed comments have helped us to improve the manuscript. Below is our point-by-point response to your comments (text in blue denotes our response).

**Comments 1:** I just wonder if the RMSEs in Figure 2c and Figure 4d are smaller than that of climatology.

**Response 1:** We appreciate your comment on the comparison between the RMSEs in Figures 2c and 4d and the climatological values. However, we are unsure if you are suggesting using climatology as the estimate for SWH overall time. If so, it is evident that the errors of climatology would be significantly larger than those of the AI model. We implemented a climatology-based model as per your comment. The results are shown in the figures below.

[Figure]

**Figure R1.** Error curves with climatology added on Figure 2c

[Figure]

**Figure R2.** RMSE distribution of the AI model (Figure 4d)

[Figure]

**Figure R3.** RMSE distribution when using the climatology as the estimates

As expected, the RMSEs are significantly higher than those from our model when using climatology as the estimate. The RMSE of the climatology remains consistently above 0.9 m throughout the error curve. In the spatial distribution of RMSE errors for the climatology model,

the errors in high-latitude regions exceed 1.5 m, showing that the climatology is not a good estimate.

**Comments 2:** For the sake of comparison, the color bar scales in Figures 6-7 should be the same as those used in Figures 4-5.

**Response 2:** We thank your suggestion to unify the color bar scales for better comparison. we have adjusted the color bar scales in Figures 6 and 7 to match those used in Figures 4 and 5. These changes enhance the clarity of how the results improve after data assimilation. The assimilation process reduces errors and increases the accuracy of the model. Notably, it is now clear that the correlation coefficient of the swell pools in the equatorial regions increases significantly with data assimilation. The updated figures will be included in the revised manuscript.

[Figure]

**Figure 4:** Comparison of SWHs from the AI model at 240-h hindcast time (when the errors are stable) with ERA5 for the year 2020. (a) Scatter plot between the SWHs from the two datasets. (b-e) Global spatial distributions of CC, bias, RMSE, and SI, respectively.

[Figure]

**Figure 5:** The same as Fig. 4, but the comparison is between the 240-h SWH hindcasts of the AI model and the CCI-Sea State dataset.

[Figure]

**Figure 6:** The same as Fig. 4, but the AI model has assimilated the data from CCI-Sea State every six hours.

[Figure]

**Figure 7:** The same as Fig. 6, but the comparison is with the CCI-Sea State dataset.

**Comments 3:** The units of variables on the vertical axis are missing in (b) and (c) in Figures 2-3.

**Response 3:** We thank you for pointing out the missing units on the vertical axis in panels (b) and (c) of Figures 2 and 3. We have added the appropriate units to the vertical axes of these panels to ensure clarity and completeness. Thank you for highlighting this oversight, as it has helped improve the accuracy and presentation of the figures. Updated figures will be incorporated in the revised manuscript.

[Figure]

**Figure 2:** The variation of global overall error metrics between the AI SWH model outputs and ERA5 with simulation time: (a) CC, (b) bias, (c) RMSE, and (d) SI. The orange and blue lines represent the mean values of the error metrics for the 236 experiments starting from different initial SWH fields, before and after assimilation, respectively. The shaded areas around the lines

indicate the range of error metrics across different experiments with varying initial SWH fields.

[Figure]

**Figure 3:** The variation of global overall error metrics between the AI SWH model outputs and ERA5 with simulation time: (a) CC, (b) bias, (c) RMSE, and (d) SI. The orange lines and shaded areas are the same as those in Fig. 2, but no epoch ensemble is used. The blue lines and shaded area are the corresponding results for the cold start with an initial field of zero SWH.

---

## Author Comment (AC2)

**Dear Chief Editor,**

Thank you for your detailed feedback and for providing clear guidance on ensuring our manuscript complies with the journal's "Code and Data Policy." We have carefully addressed the issues raised. First, we have archived our code and data in the Zenodo and assigned it and the GitHub repository an appropriate open-source license (Apache License 2.0), ensuring long-term accessibility and usability. The DOI for the Zenodo archive is 10.5281/zenodo.14244062 and can be accessed via https://zenodo.org/records/14244062.

Regarding the WW3-ST6 dataset, it is already a well-known and widely-used dataset in the wave community and is readily accessible by contacting the authors via email. However, we understand the journal's requirement for open access. To address this, we have uploaded the subset of WW3-ST6 used in this study to the Zenodo repository (10.5281/zenodo.14244062), allowing readers to reproduce the comparisons presented in the manuscript.

Our training data consists of 18 years of ERA5 reanalysis data of global wind speed and significant wave height, totaling approximately 150 GB, which is too large for most data repositories. As this dataset is publicly available through the Copernicus Climate Data Store (CDS), we provided the scripts in the aforementioned repository to assist readers in efficiently and accurately downloading the data required for the model, which should be sufficient to replicate the outputs. Besides, in the Data and Method Section, we have stated clearly that "we utilized the global SWH and 10-meter longitudinal and latitudinal components of neutral wind (U10 and V10) from the ERA5 dataset for the period 2000-2017 to train the global AI SWH model. The corresponding data in the year 2022 was used for validation to prevent over-fitting, while the model testing was conducted with data in the year 2020. Both the wind and wave data used here are at a $0.5° \times 0.5° \times 1h$ spatio-temporal resolution". Any readers who have experience downloading data from CDS should be able to download the data mentioned above. Regarding the output, the outputs generated by our rolling model are derived from over 200 different initial conditions on the test sets, exceeding 100 GB in size, which is also too large for most data repositories. To facilitate reproducibility, we have uploaded all the required test data to the Zenodo repository (10.5281/zenodo.14244062). Readers can easily reproduce the outputs by running the code provided in the repository without the need to download any additional data themselves.

Furthermore, we have included additional data in the Zenodo repository (10.5281/zenodo.14244062) to support validation and further analysis. These include the original data files used for the performance evaluation of the models based on altimeter data and the movies provided in the manuscript's supplementary materials.

We have updated the manuscript's "Open Research" section and these modifications will be reflected in the revised manuscript. It now states:

The ERA5 data is downloaded from Copernicus Climate Data (https://cds.climate.copernicus.eu/). The CCI-Sea State dataset is downloaded from the Centre for Environmental Data Analysis (https://archive.ceda.ac.uk/). The WW3-ST6 dataset is available from Liu et al. (2021), and the subset used in this study is available in the Zenodo repository (https://zenodo.org/records/14244062). The AI models established in this study and relevant test data have also been archived in the Zenodo repository (https://zenodo.org/records/14244062).

We believe these updates address the issues raised and ensure compliance with the journal's policy. Please let us know if there are any additional concerns or further clarifications required. Thank you for your time and assistance.

---

## Author Comment (AC3)

**Response to Reviewer 2:**

Dear Reviewer:

We would like to thank you for your patience in reading the paper in detail and your valuable comments. We sincerely think their detailed comments have helped us to improve the manuscript. Below, we present our point-by-point response (text in blue denotes our replies). We hope the manuscript is now acceptable following our revisions and explanations.

**Major comments:**

**Comments 1:** It is not clear what goal this AI SWH model would like to achieve. Is it to make good 0-10 days SWH predictions similar to traditional numerical wave models? If yes, we should use predicted wind fields from GFS (or IFS) as the forcing instead of the ERA5 reanalysis wind fields. In the current manuscript, one year of SWH series was generated by the AI SWH model. But it essentially functions like post-processing the ERA5 wind analysis to 0-6h short-term SWH forecasts. This is not the regular 0-10-day SWH forecasts we normally expect.

**Response 1:** Thank you for your comments. Regarding the question of whether our model is solely designed for wave prediction, the answer is "No. " While one application of our AI SWH model is wave prediction, it is essential to emphasize that wave models—whether numerical or statistical—are not limited to forecasting wave conditions over a few days. Wave models and weather forecasting models fundamentally differ in purpose and methodology. Weather forecasting is a typical initial value problem, focusing on accurate predictions of atmospheric conditions over short periods, with outputs highly sensitive to initial conditions. In contrast, phase-averaged wave models, such as Numerical Wave Models (NWMs), address a forcing problem where wave evolution is primarily driven by external forcings, such as wind fields. These wave models are used not only for forecasts but also for hindcasts and projections.

A robust wave model should perform well when driven by high-quality forcing fields and exhibit errors proportional to the quality of those fields. Therefore, wave models are typically evaluated using high-quality reanalysis forcing fields to isolate errors originating from the model itself rather than from the forcing fields. For this reason, we did not use forecasted wind fields (e.g., GFS or IFS forecasts) as forcing inputs for our model, as their larger errors could compromise the training and calibration of the wave model.

The primary goal of our study was to develop an AI-based SWH model that mirrors the input-output structure of NWMs but excludes wave spectra to reduce computational complexity. We anticipated that the absence of directional wave spectra might lead to error divergence in rolling predictions, even during hindcasts. However, the rate of divergence was unclear. Our findings demonstrate that the error stabilizes after approximately 200 hours, with stabilized errors comparable to those of state-of-the-art NWMs, particularly in wind-sea-dominated regions. This highlights the potential effectiveness of our AI-based wave model surrogate.

Traditional NWMs often downplay data assimilation's importance due to the forcing-driven nature of directional wave spectra evolution. However, the observed error divergence in our AI SWH model underscores the significance of initial conditions. Thus, the second focus of our study was to assess whether direct data assimilation of altimeter-derived SWH (bypassing wave spectra) could mitigate error divergence. Our results confirm that assimilating altimeter data effectively reduces error divergence and enhances the model's reliability.

Furthermore, the error divergence of our AI SWH model, even during hindcasts, highlights its resemblance to an initial value problem, unlike traditional NWMs. Consequently, data assimilation becomes particularly beneficial, especially in swell-dominated regions. Without data assimilation, as shown in Figure 4, the model exhibits poorer performance in "swell pools." To our knowledge, this study is the first to address error divergence in AI-based SWH rolling models and propose altimeter data assimilation as a solution. Additionally, the absence of directional wave spectra simplifies SWH assimilation by eliminating the need for spectral adjustments—an advantage not previously reported in the literature.

**Comments 2**: Section 2.4 and other relevant parts: (1) data assimilation will improve initial conditions, but it is not related to the establishment of this AI SWH model and it is suggested to remove this part from this manuscript; (2) the data assimilation method here is too simple. Nowadays, we would expect at least a 2DVAR method that considers the uncertainties of the background and the observations.

**Response 2:** We appreciate the concerns raised by the reviewers in the data assimilation section.

For comment (1), As previously mentioned, the AI SWH model developed in this study exhibits error divergence, even during hindcasts, due to the absence of directional wave spectra. This error divergence suggests that the AI SWH model shares more characteristics with an initial value problem than traditional NWMs. Consequently, data assimilation becomes an essential component of the AI model, particularly in swell-dominated regions. Without data assimilation, as demonstrated in Figure 4, the model performs poorly in "swell pools. " This represents a significant limitation. To our knowledge, this study is the first to identify and address error divergence in AI-based SWH rolling models by proposing the assimilation of altimeter data as a solution. Additionally, the absence of directional wave spectra simplifies the assimilation of SWH, eliminating the need for spectral adjustments. To the best of our knowledge, this straightforward assimilation capability of the AI SWH model has not been reported in previous studies.

For comments (2), there are two primary practical and computational reasons for our preference for objective analysis (OA)—a form of optimum interpolation (OI)—over variational methods:

1. OA is significantly less computationally demanding than variational methods. One of the AI model's greatest advantages is its efficiency and lightweight nature. Introducing variational methods for assimilation would increase computational demands by several orders of magnitude, rendering the AI model inefficient and impractical. Furthermore, OI enables incremental assimilation of observations, allowing for continuous updates as new data becomes available. In contrast,

variational methods typically require a complete assimilation cycle, which may not be feasible for fast-paced AI applications.

2. To our knowledge, there is no clear evidence that variational methods outperform OI in wave modeling. For instance, major operational centers such as ECMWF continue to use OI for wave modeling (IFS DOCUMENTATION – Cy43r1 –PART VII: ECMWF WAVE MODEL, available at: https://www.ecmwf.int/sites/default/files/elibrary/2016/79992-ifs-documentation-cy43r1-part-vii-ecmwf-wave-model_1.pdf), although there is still debate on whether NWMs need data assimilation. Therefore, our lightweight assimilation scheme is sufficient for the AI SWH model, as supported by our results.

We have also added some of the above explanation to our revised manuscript.

**Specific comments:**

Lines 48-49: "However, …" The statement is not accurate. The SWH prediction is both a forcing and an initial value problem. When focusing on the short-term (a few hours to a day) forecasts, the initial value will have a large impact. Zhou et al., 2021 and Ouyang et al., 2023 targeted 24h and 3 days forecasts respectively.

Thank you for pointing out this issue. We acknowledge that SWH prediction involves both a forcing and an initial value problem, especially when focusing on short-term forecasts. We will revise the manuscript to clarify this distinction. We will make the following changes in the revised manuscript: "Consequently, some studies have already explored AI applications in wave modeling. Some have attempted to replicate the AI weather forecasting approach by treating wave modeling as a purely nonlinear auto-regression problem of spatio-temporal series (e.g., Zhou et al., 2021; Ouyang et al., 2023). However, this approach overlooks the fact that phase-averaged wave modeling should not be treated as an initial value problem. Without a wind field driving the model, it is physically impossible to accurately simulate waves directly from past wave evolution alone. While initial conditions do play a role in short-term prediction, these auto-regression models cannot even run without the initial conditions provided by an NWM. "

Additionally, we emphasize that while initial values are essential for short-term wave forecasts, they can be readily derived from past forcing fields. This characteristic underscores why wave modeling is often viewed more as a forcing problem rather than an initial value problem.

Line 124-125: "Particularly, if a long series…" Could you clarify which part of Song and Jiang, 2023 drew this conclusion?

Song and Jiang (2023) successfully modeled the directional wave spectrum at a single point using only historical wind field data, without relying on any initial wave field conditions. Their approach was based on the premise that waves are either generated by local winds or influenced by remote historical winds. In their models, the only inputs were wind fields from the preceding 240 hours, with no wave data used as inputs, yet they achieved good results. This conclusion is further supported by a recent study by Wang and Jiang (2024) (Physics-guided deep learning for skillful wind-wave modeling, Science Advances, https://www.science.org/doi/10.1126/sciadv.adr3559). In

this study, global SWH was modeled using only wind fields from the past 240 hours, without incorporating initial SWH field data, also yielding good results.

Lines 133-134: "We believe…": Evidence, instead of a subjective "belief", is expected here to demonstrate why U-Net is suited for this work.

While there may be various reasons for selecting a specific AI model architecture, determining whether a model is genuinely well-suited to a particular problem requires thorough testing. The rationale for choosing U-Net is briefly outlined in the manuscript: "The encoder progressively extracts features from the input through convolution and pooling, while the decoder reconstructs spatial resolution using de-convolution and up-sampling. Skip connections link corresponding layers of the encoder and decoder, preserving high-resolution details". Admittedly, this initial selection was based on a subjective belief regarding the model's suitability. However, our results provide strong evidence that U-Net is indeed well-suited for this task.

To present this explanation more objectively, we have revised the text as follows: "Such a CNN-based deep learning model is well-suited for wave statistical modeling using our input-output structure, and the effectiveness of U-Net in in wave modelling has been shown in previous studies (e.g., Gao and Jiang, 2023, Wang and Jiang, 2024). "

Lines 135-136: "The processes of both…": In AI for NWP practices, lots of literature has demonstrated that AI struggles to resolve different scales at the same time (for example, smoothing to get better medium-range forecasts and hence not able to resolve smaller scales features). So how can it be assured that "The processes of both local wave generation by wind and wave propagation in space can be captured by convolutional kernels at different scales? More discussions on this are needed.

The "smoothing problem" observed in AI weather models is not an issue for AI wave models. The wave response to wind inherently acts as a low-pass filter for wind fields, as waves are essentially the integral of wind forces. This "smoothing effect" aligns well with the requirements for modeling wave generation, particularly for SWH. While for wave propagation, the propagation of swells can be regarded as a linear progress, which can also be captured be convolutional kernels at different scales. In addition, in AI weather models, the "smoothing effect" poses a significant challenge as it inhibits the model's ability to run independently without initialization from numerical models. However, this "smoothing effect" does not impact the rolling modeling of waves, as wave models, being forcing-driven, can even operate in a "cold start" mode without relying on any initial conditions.

Lines 139-142: It looks like there is a problem with the "-190 degree to 190 degree" trick: How do waves at 180 degrees propagate to -179 degrees in this method?

The method is indeed feasible, although it may not have been clearly described in the manuscript.

In the revised version, we will clarify this implementation to enhance understanding: "To handle the wraparound at the -180° and 180° longitude boundary, we employed an engineering trick of extending the input fields from -180° to 180° (720 longitudes) to -190° to 190° (760 longitudes). Specifically, the data from -180° to -170° were duplicated and appended to 180° to 190°, and a similar treatment was applied to the opposite boundary. This effectively connecting the two boundaries and avoiding discontinuities during the modeling process." This approach ensures that waves can propagate seamlessly from 180° (-180° during computation) to -179°.

Line 161-168: (1) How does the epoch ensemble method work? Need more theoretical discussions here. Generally, we would like to use all available data to train the best AI model (instead of splitting them into different epochs). (2) We can add different perturbations to generate ensembles or use ensemble wind forcings (such as from GEFS) (3) The reduction of RMSE from the 4 ensembles shown in the manuscript may come from the smoothing effect.

Thank you for your thoughtful comment. The epoch ensemble method is a commonly used engineering trick in deep learning, distinct from the concept of ensemble forecasting in NWP. Specifically, during model training, multiple models that have already converged on the validation set are saved, and their mean output is taken as the final result. Ideally, obtaining the best AI model would be preferable. However, practical limitations, such as insufficient training data, constraints in time and cost, model hyper-parameter tuning, and model complexity, often make it challenging to achieve the optimal model. This challenge is further compounded when the loss function represents the (weighted) sum of multiple components. During training, some regions may be overestimated while others are underestimated; in subsequent iterations, these regions may reverse. Achieving convergence across all locations without over-fitting can be particularly difficult. In such cases, the epoch ensemble method can provide a straightforward solution to balance these loss discrepancies.

For Comment (2), we agree that ensemble wave outputs can be generated by using ensemble wind forcings. This represents a significant potential application of our wave model surrogate. The AI model's computational efficiency allows it to process more ensemble members of wind forcing compared to NWMs, reducing time and computational costs. However, this application is beyond the scope of the discussion in this section.

Regarding Comment (3), again, we need to mention that smoothing effect is not an issue for wave modelling. Additionally, it is important to note that the epoch ensemble method involves "smoothing" across ensemble members rather than spatial or temporal smoothing of the data itself.

Fig 2: It looks like the "AI model with data assimilation" results are NOT free forecasts but analyses performed every 6 hours along with short-term (0-6h) forecasts.

Yes, the assimilation is performed every 6 hours, and the results are not forecasts but hindcasts (with ERA5 wind as inputs). Also, the time interval for assimilation can be freely configured in the provided python codes.

Line 220: "This suggests that the simple AI model can function independently, at least, in certain scenarios." What does this mean and what scenarios it refers to? Need clarifications.

Many previous AI SWH models, such as Zhou et al. (2021) and Ouyang et al. (2023), have to be used with the help of NWMs (providing the initial field), so that they cannot be used independently. While this model does not need any information from a NWM so we call it "function independently". After realizing that this might be a bit confusing, this sentence is revised to: "This suggests that the simple AI model can work without the assimilation of observation and the information from NWMs, at least, in some applications such as modelling the SWH in wind-sea dominated regions"

Fig. 3, 4, 5: Are the results in these figures from the AI model with or without data assimilation?

The results shown in Figs. 3, 4, and 5 are from the AI model without data assimilation. It is worth noting that even without assimilation, the proposed AI rolling model achieves accuracy comparable to numerical models when evaluated against the CCI altimeter dataset in wind-sea-dominated regions. To avoid potential misunderstandings, we will revise the figure captions in the revised manuscript as follows:

**Figure 3:** The variation of global overall error metrics between the AI SWH model outputs and ERA5 with simulation time: (a) CC, (b) bias, (c) RMSE, and (d) SI. The orange lines and shaded areas are the same as those in Fig. 2, but no epoch ensemble is used. The blue lines and shaded area are the corresponding results for the cold start with an initial field of zero SWH. These results do not use data assimilation.

**Figure 4:** Comparison of SWHs from the AI model (without data assimilation) at 240-h hindcast time (when the errors are stable) with ERA5 for the year 2020. (a) Scatter plot between the SWHs from the two datasets. (b-e) Global spatial distributions of CC, bias, RMSE, and SI, respectively.

**Figure 5:** The same as Fig. 4, but the comparison is between the 240-h SWH hindcasts of the AI model and the CCI-Sea State dataset.

Line 378-379: "why the AI model can slightly outperform the NWM in these areas." If we want to draw this conclusion, we will need to run both the NWM and AI models with the same settings and compare them directly.

We have to admit that it is difficult for the NWM and AI models to have the same setting because they are fundamentally different approaches. The settings used for AI models cannot be directly implemented in NWMs, and vice versa. However, we have provided a comparison between a state-of-the-art NWM hindcast dataset, WW3-ST6, and altimeter observations in Figure S4 of the Supplementary Materials. We put this figure in the Supplementary Materials to save space, because it is only used as a reference. This figure can be directly compared with Figure 5 in the main text. The results indicate that the AI model slightly outperforms the NWMs in certain regions, such as the westerlies in the Southern Hemisphere.

Lines 386-390: The initial value problem has a predictability limit (around 10 days for the wave forecasts). So one would not expect the impact of initial SWH will last beyond ~10 days.

We agree with the reviewer's observation, but it appears we may not have fully grasped the main point of this comment. Wave modeling, fundamentally, is not an initial value problem. The 10-day limit of wave forecasts primarily stems from the 10-day predictability limit of weather (wind) forecasts. If high-quality wind data are available, such as in hindcast scenarios, it is feasible to model waves over longer periods for both our AI model and NWMs.

Line 423: "An important advantage of the AI SWH model proposed here is its low computational cost compared to traditional NWMs.": It will be good if we can give concrete examples of the computational resources needed by the AI SWH model and the traditional NWMs.

We have incorporated the corresponding description into the revised manuscript according to your comment: " For example, on a personal laptop equipped with a single GTX 3080 GPU, the AI model can perform a 1-year global SWH rolling simulation at a resolution of $0.5° \times 0.5° \times 1h$ in just 10 minutes. In contrast, traditional NWMs, such as the WAVEWATCH III model, typically require several days to complete a simulation with the same output, even on supercomputing facilities."

Lines 445-446: "While training such a model would be challenging, it is not an impossible task, and the rapid advancements in AI may make this goal more achievable in the future": This sentence is too subjective, Consider revising.

We will revise the sentence to make it more objective:
"Training such a model would be challenging due to the complexity of the task, but ongoing advancements in AI methodologies, particularly in deep learning, are continuously improving the possibility of achieving this goal."

**Edit:**

Line 304: "clearly evident"  ->  "evident"

Revised.

Thank you again for your comments and suggestions. We will update the manuscript accordingly.

---

## Referee Report (RR1)

**Summary:**

This study constructed an AI model for predicting the significant wave height (SWH) parameter globally using a convolution neural network with the U-Net architecture. The AI SWH model is trained on 18 years of ERA5 reanalysis by using the SWH and the 10-m surface wind vector fields at two consecutive times (i.e., rolling prediction strategy). Therefore, the AI model "simulates" SWH in a manner similar to the numerical wave models with an initial SWH field and the forecasted 10-m wind fields. Evaluation of AI SWH model performance in 2020 shows that this AI SWH model performs as good as the WaveWatch III model with the ST6 physics. The global error patterns against ERA5 SWH and CCI-Sea State analysis product further show that the AI-SWH model produces more reliable SWH prediction in wind-sea conditions than in swell-dominant conditions. The authors conclude that this AI SWH model can be a more efficient approach to produce global forecast of significant wave height than traditional numerical wave models.

**General Comments:**

Introduction:

My impression is that the introduction somewhat overstated the powerfulness of AI model or AI SWH model. It is true that the numerical wave models have limitations in parameterizations of the wind input term and the dissipation term that govern the spectral evolutions. But I don't think the AI model are completely free from these limitations since it learns from ERA5 and inherently adopts those limitations the authors stated. I suggest the authors toning down a bit this aspect when writing about the advantages of the AI model and not giving an impression that the AI model alone could overcome the physical limitations of the numerical wave models.

Thinking about the results from a more physical perspective:

It is quite interesting that the AI model is skillful in predicting the SWH associated with wind-seas. I am just curious if this means that the AI SWH model has learned some physics of the wave evolution. Could the authors comment on whether this AI model be run in an idealized setup to produce the SWH of fetch-dependent wind waves under constant and uniform wind forcings at different wind speeds? Would the relationship between SWH and $U_{10}$ in this AI model (i.e., $(SWH-U_{10})_{AI}$) behave similar to some empirical relations between $U_{10}$, fetch, and SWH? For example, for fully developed seas, I think the authors can compute the SWH associated with the Pierson-Moskowitz spectrum at different wind speeds and obtain a SWH-$U_{10}$ relationship predicted by the Pierson-Moskowitz spectrum (i.e., $(SWH-U_{10})_{PM}$). For fetch-dependent seas similarly, $(SWH-U_{10})_{JONSWAP}$ can be found for different fetches.

**Specific Comments:**

Methods:

1. How long does it take to train this AI Model on 18 years of data? Would it be fair to mention this training time as well?
2. Would the results changes if testing was conducted using data from 2018, 2020, and 2021 together? Have the authors tested how sensitive this model is to different ratios of the training data, the evaluation data, and the model testing data? Can authors provide some answers to these questions in the method?
3. Did the authors perform some model tuning based on the evaluation dataset? if so, it would be great if the authors document what parameters have been tuned using the validation set from 2022.
4. Also, it is not very obvious to me how or why choosing 2022 for validation can prevent over-fitting. Could the authors demonstrate that this AI model is not overfitting in some way?

Results:
1. Figure 2: With data assimilation, why do the time series of the 4 error metrics have a zig-zag pattern?
2. Figure 4: Do the spatial distributions of the 4 error metrics change in different seasons?
3. By focusing on analyzing results after the errors stabilize, do the authors imply that this AI SWH model is more suitable for wave forecast beyond 10 days (240 hrs) without data assimilation and beyond 3-4 days with data assimilation?
4. Although the authors acknowledged that this paper does not compare with in-situ observations, to showcase the effectiveness of this AI model, I think it can still be worthwhile to compare the AI SWH model, WW3-ST6 hindcast, and ERA5 reanalysis, against a few in-situ buoy observations in the manner of a short time series at some key locations (e.g., some key swell-dominated locations versus wind-sea dominated locations) or weather conditions (e.g., westerlies or more uniform wind conditions versus tropical or extra-tropical cyclones).

Discussion:

It will be helpful if the authors can be more specific about the suitable applications with the AI SWH model. (e.g., time scales of the operational wave forecast, locations, seasons etc.)

---

## Referee Report (RR2)

I thank the authors for their detailed response and explanation. I mostly have minor editorial comments, which are made directly in the PDF of the manuscript (attached after this review report). Here, I summarize my main editorial comments (1-3) and list a final suggestion (4) regarding one easy analysis to strengthen the paper further. Once these comments are addressed, I think the paper will be ready for acceptance.

1. Formatting:
   - Citations in the Introduction: there are two types of format issues when citing references. a) LastName et al.**,** YYYY should be used, but sometimes the **comma** after *et al.* is missed (e.g., on line 91, Ardhuin et al. 2019 ; on line 115, Hersbach et al. 2020 ). b) sometimes, "et al" is missed for references with multiple authors. For example, on line 144 and line 149, Liu (2021) is used, but should be **Liu et al. (2021)**.
   I don't have time to mark out all the instances, but the authors need to check and fix thoroughly throughout the manuscript. It should be straightforward if the authors are using a citation tool.
   - Change U10 and V10 to $U_{10}$ and $V_{10}$, which are more commonly seen in the literature.

2. Organize information presented in the Results section better.
   - Breaking the Result section into subsections to help readers follow better: I added some suggestions directly on the PDF.
   - For Fig. 2, I think it is easier to describe results with and without data assimilation in two consecutive paragraphs, without jumping back and forth between Fig. 2 and Fig. 3. So, I suggested moving the paragraph starting on line 295 to after line 281.

3. Reflect the response to some of my previous questions/comments in a concise sentence in the manuscript wherever the authors think applicable:
   I appreciate that the authors did some extra testing to address my comments. While I agree that they do not need to include the figures attached in the response, it would be nice to add a concise sentence in the related section to show and assure readers that they have done robust testing. For example:
   - Regarding my question about choosing 2020 as the test year, I suggest the authors adding *"Note that evaluating against other untrained years yields similar results, with differences in correlation coefficient (CC) and root mean square error (RMSE) being less than 0.003 and 0.03, respectively."*, perhaps at the end of Line 268.

- For other questions, I leave it to the authors to decide whether adding extra explanation to readers in one sentence would help improve the understanding/credibility of the paper.

4. Regarding the attribution of the larger CC errors in the AI-model to swell pools:

I think the authors can support this interpretation further (beyond citing references) by showing (or binning) the error metrics as a function of the misalignment angle between wind and dominant or mean waves. Following the authors' argument, we could expect that on average, the CC error metric reduce with this misalignment angle between wind and waves (absolute value from 0° to 180°). Note that 0-45 is aligned/wind seas, 45-135 is cross-seas, and >135 is swell opposing wind at least commonly used in the hurricane-wave community (Holthuijsen et al. 2012; **https://doi.org/10.1029/2012JC007983**). I think this should be easy to do, since the mean wave direction can be obtained from ERA5 (and likely also from CCI-Sea State dataset).

I think adding this information in a figure will strengthen this paper. In fact, one concern I still have with the current version of the manuscript is that the results section is populated by a bunch of figures in the same manner simply for different data sources or with/without data assimilation (Fig. 4-9). (It is a bit excessive such that it seems Fig 7 and 8 can also be supplementary information.)

If the key message of Fig 7 and Fig 8 is just that data assimilation reduces the overall errors, especially in the swell-dominated regions. It could probably also be summarized by the extra figure I suggested above (i.e., plotting the error metrics as a function of the misalignment angle between wind and dominant or mean waves). The authors would also need to keep in mind my comment #2 above when adding this figure. The order of information in the Results section may need to be slightly re-arranged if this extra figure is included.

[revised manuscript text omitted]

Furthermore, in order to further assess the performance of the model more independently and objectively, we compared its

405 results with independent in situ observations from NDBC buoys, with the results shown in Fig.6. Similar to the comparison with altimeter data, most points align well along the 1:1 line, with bias, RMSE, CC, and SI values of -0.002 m, 0.306 m, 0.959, and 0.161, respectively. For reference, Fig. S6 in the SI presents the comparison between WW3-ST6 and NDBC data, where

the corresponding bias, RMSE, CC, and SI values are -0.004 m, 0.291 m, 0.963, and 0.153. These results are largely consistent with the comparison against CCI-Sea State data, further reinforcing the reliability of AI model.

410

[Figure]

**Figure 6:** The same as Fig. 4, but the comparison is between the 240-h SWH hindcasts of the AI model and NDBC Buoy dataset.

**3.2.2 AI model performance with data assimilation (just a vague suggestion)**

415 For the 240-h rolling hindcast results of the AI model after data assimilation every six hours, the corresponding comparisons with ERA5 and CCI-Sea State are shown in Figs. 7 and 8, respectively. Compared to the results without assimilation in Fig. 4, all the error metrics of the model improve significantly after data assimilation in Fig. 7. Specifically, the CCs in the Pacific, Atlantic, and Indian Ocean swell pools increase from ~0.7, ~0.8, and ~0.85 in Fig. 4 to ~0.88, ~0.9, and 0.95 in Fig. 7, respectively. The magnitudes of bias, RMSE, and SI also decrease across the oceans after assimilation, although the bias and

420 RMSE remain relatively high in regions to the southwest of South America and southeast of Africa. The comparison between Fig. 5 and Fig. 8 shows a similar result: the overall errors become significantly smaller, particularly in the swell-dominated regions, after assimilation. Similar to Supplementary Movie S1, the comparison animation of the results after assimilation is placed in Supplementary Movie S2, where the AI model better captured the SWH evolution. To save space, the comparison between NDBC buoy data and the AI model output after data assimilation is shown in Fig. S7 in the SI, where improvements

430 in all error metrics (bias = 0.033 m, RMSE = 0.279 m, SI = 0.147, CC = 0.967) compared to those in Fig. 6 can also be observed, showing the effectiveness of the assimilation. As a reference, the comparison between NDBC buoy data and the ERA5 is also shown in the SI (Fig. S8, with bias = 0.027 m, RMSE = 0.266 m, SI = 0.140, CC = 0.971), which is still slightly better than the results of AI model.

[revised manuscript text omitted]

---

## Author Response (AR2)

**Response Letter**

Dear Editor,

Thank you for concerning our manuscript in GMD. We appreciate you and the reviewers for your earnest work. The comments from the reviewers are very helpful, and the paper has been revised carefully according to these comments. Our point-by-point responses to the comments of the reviewers are attached, as well as a tracked-changes version of the manuscript. For the comments that we do not completely agree on (only a few), we also give our explanations in the point-by-point responses.

We hope that this version of the manuscript is acceptable for publication in GMD.

If you have any questions, please feel free to contact us. We appreciate your support very much.

Thank you for your time and consideration. We look forward to your positive response.

Sincerely,
Haoyu Jiang, Ph.D.
College of Life Science and Oceanography, Shenzhen University
Email: Haoyujiang@szu.edu.cn

**Response to Reviewer 3:**

*For the record, this manuscript was sent to me for review as a revision. This is the first time I have seen and reviewed the manuscript.*

*The authors present a data driven global wave model where a present wave height and a wind speed 6h ahead in time are used to estimate wave height 6h in the future. A "rolling model' is created by repeating this step, and data assimilation is added to make the model more accurate. The only thing in these sections that should be highlighted more is how the data assimilation is performed. On line 185 it is stated that Data Assimilation is used to "correct the model's "initial" SWH field.", whereas later (line 366) it is stated that data is assimilated "every 6 hours". Before going into more detailed critique, it needs to be stated that resulting model and its analysis are clearly suitable for publication in GMD.*

Dear Reviewer:

We would like to thank you for dedicating time to carefully read our manuscript and provide feedback. We sincerely think your detailed comments have helped us to improve the manuscript, and revisions are made according to them. A revised version of the manuscript with changes highlighted is also attached to this response letter. We hope that the revised version of the manuscript meets your expectations. For the few comments where we hold a different perspective, we have provided detailed explanations in the following point-by-point responses.

We would like to clarify that our model is a rolling forecast model with the simplest 1-hour-by-1-hour time steps. Specifically, the model takes the SWH field at time $T_i$ and the wind field at $T_{i+1}$ as inputs to predict the SWH at $T_{i+1}$, which is similar to numerical wave models (NWMs). We believe this has been clarified in Section 2.2.1. In this setting, the outputs of the last time step will be the "initial" field of the next time step, and the frequency of data assimilation can

be user-defined. Here, assimilation was conducted every 6 hour in our experiment, but a higher or lower assimilation frequency can also be used. Generally, a higher assimilation frequency leads to more accurate results but also entails increased computational costs, and vice versa.

To better clarify this, we added some explanation to the revised manuscript:

"…we tried to incorporate data assimilation techniques by integrating altimeter measurements to correct the model's "initial" SWH field. It is noted that in our input-output setting, the outputs of the last time step will be the "initial" SWH field of the next time step…."

"In our data assimilation experiment, assimilation was conducted every six hours (i.e., every six time steps, observations are used to corrected the outputs of the rolling model and the updated outputs are used as the new inputs at the next time step), beginning after the first 24 hours of the model run.  Of course, the frequency of data assimilation can be user-defined. A higher assimilation frequency generally leads to more accurate results but also entails increased computational costs, and vice versa."

*The input and target of the study is the ERA5 reanalysis. Note that this reanalysis consists of a wave hindcast with most altimeter data assimilated into it. Note that due to the lack of sufficient wave data to generate a data dominated wave analysis, ERA5 still is mostly a wave model hindcast, and that its quality is inhomogeneous. As ERA5 is more accurate at the locations of the assimilated altimeter data, validating with ERA5 and cross referencing with the same altimeter data is a little incestuous and produces error measures that are too rosy. For this reason, I would have preferred developing the AI model with hindcast without DA, and then comparing the AI model, the input model and the altimeter data in a three-way approach would be a cleaner analysis of various data sources. This is not disqualifying for the present study, but the limitation (validation with dependent data) needs to be discussed.*

We acknowledge the problems of ERA5 that you mentioned. In spite of these problems of ERA5, training an AI model requires high-quality input data to achieve reliable results. From this perspective, ERA5 is still a good dataset to train against.

Regarding the independence of result evaluation, we respectfully disagree the comment that "validating with ERA5 and cross referencing with the same altimeter data is a little incestuous and produces error measures that are too rosy". If the comparison is made between ERA5 and the altimeter data that has been assimilated to ERA5, it will be, of course, incestuous or even unreasonable. However, the comparison is made between the AI model and ERA5, and between the AI model and altimeter measurements. We need to emphasize that once the training of the AI model is finished, it can be regarded as a model logically independent of ERA5 (or any hindcast dataset it is trained against). We can draw an analogy between the training process of AI models and the tuning process of NWMs, as both are essentially adjusting a set of empirical coefficients (though AI models typically have far more parameters to "train"). Specifically, we can tune the NWMs using ERA5 reanalysis data or directly with altimeter observations, and then validate the NWM results against ERA5 reanalysis or altimeter observations in a different periods (e.g., altimeter measurements from years not used in tuning). Here, the training and validation process of our AI model strictly follows this same logic. Therefore, we believe it will not introduce problems when using the ERA5 data as the training target and testing benchmark (of course, the training set and testing set should be separated), and the comparison made in this study is totally reasonable and will not generate rosy error metrics.

Certainly, we by no means suggest that ERA5 constitutes the optimal training dataset for developing such AI models. On one hand, there undoubtedly exist better methodologies for calibrating NWM hindcast or assimilating/merging observations into NWM hindcasts. On the other hand, NWMs themselves are continually evolving, with their output data achieving progressively higher accuracy. The more fundamental objective of this manuscript remains

investigating the feasibility of this "simplest" input-output strategy in AI modeling. The quality of ERA5, according to our results, is adequate for this purpose.

We hope this explanation meets with your approval. Should you hold differing views, we warmly welcome your further comments in the review report and would be pleased to continue the discussion on this matter.

*As a traditional wave modeler who has also worked in AI for decades, I find the justification for doing an AI model weak. On line 9 it is stated that wave models "are computationally intensive and constrained by incomplete physical representations of wave spectral evolution.", yet the ERA5 data used here is founded in these limited models. Moreover, wave models provide much more data than the SWH for practical wave predictions, or in coupled environmental models. Yes, NWM are more expensive, but we have been providing operational forecasts since the 1960s with such models (contrary to suggestions provided on line 36), so apparently the expenses are not prohibitive. The paper will be stronger without half-baked justifications.*

We completely agree with the reviewer that NWM also has many (more) advantages compared to contemporary AI wave models (including the one in this study). Here, we list the limitations of NWM merely to demonstrate that the proposed AI model still has its merits that can overcome some of the NWM's problems, not to claim that the AI model outperforms NWM, let alone suggest it could replace NWM.

Although the ecWAM which ERA5 is based on also has the problem of incomplete physical representations of wave spectral evolution and numerical effects (e.g., discrete interaction approximation and garden sprinkler effect), these effects can be alleviated if enough data is assimilated/merged to the model output and assimilation can generate a more reliable analysis field. This is why today's AI weather forecasts nowadays can beat numerical ones in some error

metrics by training against ERA5. Using the data combining observations and NWM outputs, we also realized a AI model that has better accuracy than NWMs with respect to SWHs recently (with a different input-output strategy) (Wang and Jiang 2024). However, we have to admit that the aim of this manuscript is not to overcome the incomplete physical representations and numerical effects, so we just simply mention this problem in the introduction.

To provide a balanced perspective on both NWMs and AI-based wave models, we have added clarifying statements in the introduction's concluding section to address your concerns:

"Although good results have been obtained by the AI model presented in this study, it is noted that we do not intend to suggest that the AI model is superior to traditional NWMs or that it could replace NWMs. NWMs still retain numerous advantages over AI approaches, such as their ability to provide parameters beyond SWH and their stronger physical interpretability, among other merits. The AI model we have developed should be more regarded as a model surrogate specifically for time- or computation-sensitive scenarios."

**Ref.:**
Wang, X., & Jiang, H. (2024). Physics-guided deep learning for skillful wind-wave modeling. Science Advances, 10(49), eadr3559.

*Considering that wind waves dominated the higher wave heights and the overall errors worldwide, a model like this AI model that is focused on representing wind seas should result in reasonable wave heights but also will have issues in areas with multiple (dominant) swell fields. This is acknowledged in the manuscript in discussing the errors in "swell pools". It would be nice to acknowledge the need of being able to do swell accurately too for many applications. Since the "waves across the Pacific" studies in the 1960s, it is well known that NWMs can do this well. Note that with the dominance of wind seas in model errors, the differences between cold and hot started results, as well as impacts of DA are at least*

*qualitatively expected. Note that this could be expected as present wave height and future wind speed allow for assessing where the dominant wind wave field is with respect to growth stage of the wave field. Did you check if additional accuracy can be attained if the present wind field is used too (this would be a proxy for wind sea and swell separation in the initial wave height in the algorithm)?*

As noted, the model performs well in wind-sea conditions but less so in swell regions, as discussed in the "swell pools" error analysis. Undeniably, swells play a crucial role in many applications, thus, there are needs of being able to model swell accurately. However, although the propagation of swells has been well understood since the famous field experiment, 'Waves Across the Pacific', they their behavior retains characteristics of an initial-value problem, making swells a persistent source of error in current NWMs (e.g., Jiang et al. 2016). Therefore, NWM also usually perform worse in swell-dominated regions than in wind-sea-dominated regions. From Figure S4 in the Supporting Information, it can be seen that NWM demonstrates similarly low correlation coefficients in "swell pool" regions, which is only marginally higher than our AI model (as shown in Figure R1 below). This explains why we consider the AI model's performance in swell-dominated regions is still acceptable.

Regarding the suggestion to incorporate the present wind field as an additional input, we made some tests according to the reviewers suggestion. However, we found that it did not lead to improvement compared to the current model. This is not surprising because the variation of wind is usually very small within one hour so that the pattern of the present wind field and the 1-h future wind field is very similar. Moreover, although wind-sea and swell SWH can be roughly separated using wind speed + SWH information (which we believe is exactly the reason for our AI model can still model swells with acceptable accuracy), the evolution of swell SWH is also dependent on the direction and period of swells.

[Figure]

***Figure R1.*** *The spatial distributions of correlation coefficients (upper) and RMSE (lower) between model results and CCI-sea state data in 2020 for global ocean: (left) AI model V.S. CCI-sea state, and (right) WW3-ST6 NWM V.S. CCI-sea state. The left column is from Figure 5 in the manuscript and the right column is from Figure S4 in the Supporting Information.*

While adding the present wind field is insufficient for swell modelling, adding additional historical wind fields can indeed improve swell modelling, as demonstrated in our previous paper (Wang and Jiang 2024). This enhancement stems from the nonlinear teleconnection between swell energy and distant historical wind forcing. However, with increased wind field inputs, the significance of the initial SWH field diminishes substantially, while the model's physical framework and input-output relationships undergo fundamental modifications. Consequently, in the present study, we maintain our methodological focus on the simplest rolling modelling approach, i.e., one wind field + one SWH field.

**Ref.:**

Jiang, H., Babanin, A. V., & Chen, G. (2016). Event-based validation of swell arrival time. Journal of Physical Oceanography, 46(12), 3563-3569.

Wang, X., & Jiang, H. (2024). Physics-guided deep learning for skillful wind-wave modeling. Science Advances, 10(49), eadr3559.

*Specific comments:*

*Line 17: "… the errors of the model diverge lightly …". Perhaps use "accumulate" or "increase" as this is not divergence in the classical meaning in environmental sciences.*

Thank you to the reviewer for pointing this out and we have changed the words from "diverge/divergence" to "accumulate/accumulation" in the revised manuscript.

*Line 20: "This deep learning model can not only serve as an efficient surrogate for traditional numerical wave models but also provide a baseline for statistical modeling of global SWH due to its simplicity in inputs and outputs." For model uncertainty, where generally only wave heights are considered, this indeed could be a good application.*

We appreciate your positive recognition of the potential application of our deep learning model in providing a baseline for statistical modeling of global SWH. To better stress this model is only for wave height, this sentence is slightly revised to :

"This deep learning model can not only serve as an efficient surrogate for traditional numerical wave models with respect to SWH but also…"

*Line 32-33: WW3 is referred to by its manual, SWAN by foundational papers. Please balance your references (I would prefer foundational references, and manuals only when the model is used to identify the version).*

Thank you for pointing this out and we have changed the citation:

Tolman, H. L.: The numerical model WAVEWATCH: a third generation model for hindcasting of wind waves on tides in shelf seas, Delft University of Technology, Department of Civil Engineering, Fluid Mechanics Group, Delft, 1989.

*Line 114: The WW3 data is not used here at all. Why is it in the materials section then?*

We did compare the performance of the AI model with the WW3-ST6 in the part of discussing the error of 'swell pools'. This is why we can say that our model is comparable to those of state-of-the-art NWMs, and why we can say AI model performs well across global oceans in general, both in wind-sea- and swell-dominated regions (although the expression "both in…" is removed in the revised manuscript). To save the number of figures in the text and to make the manuscript more reader-friendly, we put the figures of the results of the comparison between the WW3-ST6 and the CCI-Sea State in the Supporting Information Figure S4.

*Line 185: DA use for "initial" wave height is misleading, as data is assimilated every 6h (Line 366).*

Thank you for pointing this out. To better explain this, we added a sentence after this part: "It is noted that in our input-output setting, the outputs of the last time step will be the "initial" SWH field of the next time step." This should eliminate the potential misunderstandings.

*Line 193: Adding a measure for the representation of the signal such as variance of the wave height (as in a Taylor diagram) would be useful, since minimizing a rms error tends to result in a smooth model that smooths out some highs and lows.*

We are a bit confused about this comment as Line 193 (equations) seems to be not relevant to your comment. We assume you are talking about the error metrics. However, when the RMSE and correlation coefficient are known, the location in the Taylor diagram is determined.

Therefore, we think the four error metrics used in this study is sufficient to describe the error property. In our opinion, Taylor diagram is more suited for the comparison of several different models, but here we only have AI model and WW3-ST6, thus, we do not feel the necessity of using a Taylor diagram.

Also, minimizing the RMSE does not necessarily tend to result in a model that smooth out highs and lows. This only happens only when the problem is too complex for the model to accurately capture the high and low variations. When it comes to the modelling of SWH, we believe this poses even less of a concern because the variation of SWH is usually smooth itself since SWH can be regarded as a "low-pass filter" of winds.

If we misunderstood this comment, it will be nice if you can expand it a bit.

*Figure 2: It appears from the wavy behavior in the DA runs that data is created every 3 hours, not every 6 hours are claimed in the description of the AI model. This needs to be explained before publication.*

We confirm that data assimilation is performed every 6 hours, as described in the manuscript, and this can also be verified in Figure 2. For example, in Figure 2(a), the red box highlights a 24-hour period. Within this period, the blue curve exhibits 4 distinct upward steps, each corresponding to an assimilation event. These steps indicate the corrections applied to the model every 6 hours. We hope this clarification addresses any concerns about the assimilation frequency.

[Figure]

***Figure R2.*** *Figure 2a in the manuscript. To better show the wavy behaviour of the curve, we use a red rectangular to show the four upward steps within 24 hours. The lines represent the mean values of the error metrics for the experiments starting from different initial SWH fields. The shaded areas around the lines indicate the range of error metrics across different experiments with varying initial SWH fields.*

***Line 250: Note that the general statement about the effects of DA is accurate, but does not acknowledge that validating with the same but sparse data is not likely to be representative for areas without a recent observation.***

We are also a bit confused about this comment. Are you suggesting that sparse altimeter observations are insufficient for evaluating areas without a recent observation? However, it should be emphasized that these results derive from 236 parallel experiments, in which the altimeter data can be considered effectively global in coverage. Moreover, most altimeters complete an Earth orbit in under two hours, resulting in spatial dislocations between the validation dataset and previously assimilated altimeter observations. To further address this issue, we included independent in-situ buoy observations for further validation (Figure 6 and S7 in the revised manuscript). The results of the comparison are in good agreement with those obtained from the altimeter data, supporting the robustness of our model.

Again, if we misunderstood this comment, it will be nice if you can expand it a bit.

*Line 294: many years of experience with the GSE has taught us that the GSE results in unrealistic wave fields but has little impact on error statistics. I find this argument weak for that reason.*

We respectfully maintain a differing perspective: when wave fields appear unrealistic, this indicates that data errors/inconsistencies have become substantial enough to be visually identifiable. In such cases, maybe one should no claim that "the error statistics have little impact on error statistics". In many scenarios, since SWHs in swell-dominated regions are typically smaller, the absolute errors (e.g., RMSE) induced by the GSE may appear small. However, this often corresponds to significantly increased relative errors, manifested through decreased correlation coefficients. While the random errors from GSE may not be as consequential as the first two factors mentioned, we contend they nevertheless make non-negligible contributions to the overall error.

*Lines 326-333: The statements on the first and last lines about swell and wind seas do not seem to be consistent.*

These two statements are not contradictory. The first statement conveys that the AI model demonstrates reasonably robust performance across both wind-sea and swell conditions. The last statement indicates that even if readers consider the swell performance less optimal, they should at least acknowledge the model's capability in wind-sea scenarios—where it could serve as an effective surrogate for NWMs.

To clarify this point and prevent potential misinterpretation, we have further removed the phrase "both in wind-sea- and swell-dominated regions."

*Summary:*

*This study constructed an AI model for predicting the significant wave height (SWH) parameter globally using a convolution neural network with the U-Net architecture. The AI SWH model is trained on 18 years of ERA5 reanalysis by using the SWH and the 10-m surface wind vector fields at two consecu6ve 6mes (i.e., rolling prediction strategy). Therefore, the AI model "simulates" SWH in a manner similar to the numerical wave models with an initial SWH field and the forecasted 10-m wind fields. Evaluation of AI SWH model performance in 2020 shows that this AI SWH model performs as good as the WaveWatch III model with the ST6 physics. The global error patterns against ERA5 SWH and CCI-Sea State analysis product further show that the AI-SWH model produces more reliable SWH prediction in wind-sea conditions than in swell-dominant conditions. The authors conclude that this AI SWH model can be a more efficient approach to produce global forecast of significant wave height than traditional numerical wave models.*

Dear Reviewer:

We would like to thank you for your patience in reading the paper in detail and your valuable comments. We sincerely think your detailed comments have helped us to improve the manuscript. Below, we present our point-by-point response (text in black denotes our replies). We hope the manuscript is now acceptable following our revisions and explanations.

*Major comments:*
*Introduction:*
*My impression is that the introduction somewhat overstated the powerfulness of AI model or AI SWH model. It is true that the numerical wave models have limitations in*

*parameterizations of the wind input term and the dissipation term that govern the spectral evolutions. But I don't think the AI model are completely free from these limitations since it learns from ERA5 and inherently adopts those limitations the authors stated. I suggest the authors toning down a bit this aspect when writing about the advantages of the AI model and not giving an impression that the AI model alone could overcome the physical limitations of the numerical wave models.*

We completely agree with the reviewer that numerical wave models (NWMs) also has many (more) advantages compared to contemporary AI wave models (including the one in this study). Here, we list the limitations of NWM merely to demonstrate that the proposed AI model still has its merits that can overcome some of the NWM's problems/limitations, not to claim that the AI model outperforms NWM, let alone suggest it could replace NWM.

To provide a balanced perspective on both NWMs and AI-based wave models, we have added clarifying statements in the introduction's concluding section to address your concerns:

"Although good results have been obtained by the AI model presented in this study, it is noted that we do not intend to suggest that the AI model is superior to traditional NWMs or that it could replace NWMs. NWMs still retain numerous advantages over AI approaches, such as their ability to provide parameters beyond SWH and their stronger physical interpretability, among other merits. The AI model we have developed should be more regarded as a model surrogate specifically for time- or computation-sensitive scenarios."

*Thinking about the results from a more physical perspective:*
*It is quite interesting that the AI model is skilful in predicting the SWH associated with wind seas. I am just curious if this means that the AI SWH model has learned some physics of the wave evolution. Could the authors comment on whether this AI model be run in an idealized*

*setup to produce the SWH of fetch-dependent wind waves under constant and uniform wind forcings at different wind speeds? Would the relationship between SWH and U10 in this AI model (i.e., (SWH-U10)AI) behave similar to some empirical relations between U10, fetch, and SWH? For example, for fully developed seas, I think the authors can compute the SWH associated with the Pierson-Moskowitz spectrum at different wind speeds and obtain a SWH-U10 relationship predicted by the Pierson-Moskowitz spectrum (i.e., (SWH-U10)PM). For fetch dependent seas similarly, (SWH-U10)JONSWAP can be found for different fetches.*

Thank you for this insightful comment.

The AI model is trained to learn the statistical relationships present in the training dataset rather than explicitly solving physical equations governing wave evolution. From this perspective, one can say that the AI model has learned some physics of the wave evolution, from a statistical point of view. In particular, our cold-start experiments demonstrate that when driven by realistic wind fields, the AI model progressively produces results closer to ground truth. This suggests, to some extent, that the AI has learned quantitative patterns of wave growth (labelling these as physics may be inappropriate—what the AI discovers remains fundamentally statistical in nature). Furthermore, SWH data in current wave models/reanalyses fundamentally adhere to the statistical relationships between U10 and SWH. When the AI model produces results consistent with these wave models/reanalyses, it implicitly indicates that these established relationships are largely preserved within the AI model.

However, we need to note that the AI model works in a different way from NWMs. After the AI model finish its training, the input form of the AI model needs to be exactly the same as that used in the training. In our case, the input of the our AI model has to be global SWH at $T_i$ and global wind field at $T_{i+1}$. Some important but constant information, such as bathymetry and coastal morphology, and even the curvature of the Earth, are implicitly embedded in the AI model in a statistical way. Therefore, from our understanding, it seems to be impossible to

conduct idealized tests that can be easily done by NWMs, such as fetch-limited and duration-limited tests, in our AI model framework. This is also a limitation of the AI model. In our AI model, given the global domain of simulation, even prescribing a spatially "uniform" wind direction in the input fields is inherently unfeasible due to the spherical effect. Similarly, the AI model is not suitable for certain toy model experiments, e.g., we cannot setup a simulation of global SWH on an Earth without any land, which NWMs can easily handle.

***Specific comments:***

***Methods:***

***1. How long does it take to train this AI Model on 18 years of data? Would it be fair to mention this training time as well?***

We have mentioned the training time in Section 2.2.3 (Model Training) in the revised manuscript, which reads: "We used six batches for training and trained the model for up to 30 epochs at a learning rate of 0.0001 using the AdamW optimizer. To alleviate overfitting, we implemented a commonly used deep learning technique where training is halted when the loss in the validation set does not decrease for four epochs. Using our training samples (data from 2000 to 2017), training took approximately one hour per epoch on an NVIDIA RTX 4090 GPU. " Therefore, it takes less than ~30 hours to train this AI model using the data from 2000 to 2017.

***2. Would the results changes if testing was conducted using data from 2018, 2020, and 2021 together? Have the authors tested how sensitive this model is to different ratios of the training data, the evaluation data, and the model testing data? Can authors provide some answers to these questions in the method?***

According to the your suggestion, we also conducted the model test using the data from 2018, 2019, 2020, 2021. The following Figure R3 shows the presents the error curves of the AI model

on the test sets from 2018, 2019, 2020, and 2021. From this figure, it is evident that the model performs consistently across these three test sets, with differences in correlation coefficient (CC) and root mean square error (RMSE) being less than 0.003 and 0.03, respectively. Such differences are similar to the difference of error metrics for NWMs across different years. This results demonstrate that the model exhibits strong robustness and generalization ability across different test periods. This results align with our expectations: a properly developed statistical model, when evaluated on unseen test data, should demonstrate consistent performance across different years.

[Figure]

***Figure R3.*** *The variation of global overall error metrics between the AI SWH model (training with the data of years 2000-2017) outputs and ERA5 with simulation time using data of different years as the testing set: (a) CC, (b) RMSE. The lines represent the mean values of the error metrics for the experiments starting from different initial SWH fields. The shaded areas around the lines indicate the range of error metrics across different experiments with varying initial SWH fields.*

Similarly, following your comments, we tested how sensitive this model is to different amount of the training data. According to the basic knowledge of deep learning, the ratio among the three datasets is not critical. However, insufficient training data volume may indeed lead to either overfitting or underfitting issues. Figure R4 shows the error curves of the AI model trained with different amounts of training data and evaluated on the 2020 test set. It can be seen that the model performance increase with the increase of the size of the training dataset.

[Figure]

***Figure R4.*** *The variation of global overall error metrics between the AI SWH model outputs and ERA5 with simulation time using data of different periods as the training set (testing using the data of year 2020): (a) CC, (b) RMSE. The lines represent the mean values of the error metrics for the experiments starting from different initial SWH fields. The shaded areas around the lines indicate the range of error metrics across different experiments with varying initial SWH fields.*

The results presented in both Figure R3 and Figure R4 are within expectations. We have not included the these findings and their associated discussion in the revised manuscript because we intentionally avoid delving into how technical details of the AI models (e.g., training data volume, model architecture, hyperparameters, number of layers and parameters) influence the model results. To some extent, when given the predefined input-output framework, there will almost always be opportunities - however marginal - for performance improvement of AI models through such technical refinements. We are not saying these technical details lack importance, but these details primarily represent engineering challenges in model implementation: through extensive experimentation, one could systematically explore which specific model architectures, hyperparameters, and training datasets might yield optimal results within this input-output framework.

*3. Did the authors perform some model tuning based on the evaluation dataset? if so, it would be great if the authors document what parameters have been tuned using the validation set from 2022.*

No, we did not perform any model parameter tuning based on the validation/testing set.

*4. Also, it is not very obvious to me how or why choosing 2022 for validation can prevent overfitting. Could the authors demonstrate that this AI model is not overfitting in some way?*

We employed the commonly used early stopping strategy in deep learning to alleviate overfitting. Specifically, during training, we monitored the mean squared error (MSE) on the validation set, and if the MSE remained unchanged or started to increase for several consecutive epochs, training was terminated to alleviate overfitting to the training data. We realized that we forgot to mention this in the manuscript, which is our problem, this has been added to manuscript:

"We used six batches for training and trained the model for up to 30 epochs at a learning rate of 0.0001 using the AdamW optimizer. To alleviate overfitting, we implemented a commonly used deep learning technique where training is halted when the loss in the validation set does not decrease for four epochs."

In principle, the choice of a validation set is flexible as long as it does not overlap with the training or test sets. Our decision to use 2022 as the validation set was based on two key considerations: (1) The year 2022 is temporally distant from the test set, allowing for a more objective assessment of the model's generalization ability and helping to reveal potential overfitting issues. (2) There is no corresponding CCI altimeter data for 2022, meaning that this year's data could not be used for other parts of our analysis (comparing with CCI data). This made it a natural choice as an independent validation set.

Even with the implementation of early stopping strategy, we cannot conclusively demonstrate that the AI model is not overfitting. Your comments rightly reminded us that replacing "prevent" with "alleviate" would constitute a more precise formulation.

*Results:*

*1. Figure 2: With data assimilation, why do the time series of the 4 error metrics have a zigzag pattern?*

This pattern is simply a natural consequence of the data assimilation process, which is performed every 6 hours, using altimeter observations to correct the SWH output of the last time step and using the corrected SWHs as the input of the next time step. After each assimilation, the accuracy of the model output improves as errors are corrected. Then as rolling model continues, small errors continue to accumulate, leading to a gradual decline in accuracy until the next assimilation step.

*2. Figure 4: Do the spatial distributions of the 4 error metrics change in different seasons?*

Sure, the spatial distributions of error metrics will change in different seasons, because the wave climates are different for different seasons. Such a change can be observed in all wave models, such as simple statistical models, AI models, and NWMs.

Figure R5 illustrates the error curves of the AI model in different seasons during the rolling inference process. The results clearly indicate seasonal differences in errors: the overall errors are lowest in JJA and highest in DJF, although the difference is generally small.

[Figure]

***Figure R5.*** *The variation of global overall error metrics between the AI SWH model outputs and ERA5 with simulation time in different seasons: (a) CC, (b) bias, (c) RMSE, and (d) SI. The lines represent the mean values of the error metrics for the experiments starting from different initial SWH fields. The shaded areas around the lines indicate the range of error metrics across different experiments with varying initial SWH fields.*

We also examined the spatial distribution of the AI model's 240-hour hindcast errors, as shown in Figure R6. The error patterns are also different in different seasons, which is linked to the strong seasonal variations in wave climate. However, these patterns all show that the model perform well in wind-sea dominated regions and the performance degrades in swell-dominated regions.

[Figure]

**Figure R6.** *Global distributions of (left) correlation coefficients and (right) RMSEs between AI model outputs and ERA5 for different seasons: (a,b) DJF, (c,d) MAM, (e,f) JJA, (g,h) SON.*

*3. By focusing on analysing results after the errors stabilize, do the authors imply that this AI SWH model is more suitable for wave forecast beyond 10 days (240 hrs) without data assimilation and beyond 3-4 days with data assimilation?*

No, we are not implying that the AI SWH model is more reliable when the errors become stable.

It is noted that while one application of our AI SWH model is SWH forecast, it is essential to emphasize that wave models—whether numerical or statistical—are not limited to forecasting wave conditions over a few days. Besides, the performance of wave forecasting, rely not only on the performance of wave model, but also on the accuracy of wind forecasting.

Here, the reason for focusing on the results after the errors stabilize is to demonstrate that the error of the AI rolling model does not accumulate indefinitely if the model is driven by high-quality forcing fields. More importantly, after reaching a stable state, the AI model achieves accuracy comparable to state-of-the-art numerical wave models, demonstrate its usability. With available observational data, data assimilation helps the model stabilize more quickly and achieve better results. This results indicate that such an AI wave model can be used for long-term hindcasts, projections, and rolling forecasts of SWH.

For real-world forecast/hindcast problem, it is noted there cannot be "perfect" initial field. The initial field for each forecast cycle derives from either the prior forecast field or analysis field. Consequently, during rolling forecasts, the model's initial conditions at every time step typically reach a stable error state. Given these initial fields, if the AI model driven by high-quality future wind fields (e.g., analyzed wind fields), the model would maintain stable error characteristics – this is the rationale for SWH hindcast, both for our AI model and NWMs. However, in operational wave forecasting, the driving wind fields themselves accumulate increasing errors with forecast lead time, inevitably leading to progressive degradation of wave forecast quality - an inherent limitation for all wave forecasting, also for both AI models or NWMs.

*4. Although the authors acknowledged that this paper does not compare with in-situ observations, to showcase the effectiveness of this AI model, I think it can still be worthwhile to compare the AI SWH model, WW3-ST6 hindcast, and ERA5 reanalysis, against a few in-situ buoy observations in the manner of a short time series at some key locations (e.g., some key swell-dominated locations versus wind-sea dominated locations) or weather conditions (e.g., westerlies or more uniform wind conditions versus tropical or extra-tropical cyclones).*

We sincerely appreciate the reviewer's valuable suggestion. In response, we have conducted a comparison between different models and NDBC buoy observations, following the same evaluation method used for CCI-sea state altimeter data. The results demonstrate consistency

with the comparisons using CCI-sea state altimeter data. For the AI rolling model without data assimilation, once the simulation stabilizes after approximately 240 hours of rolling inference, its performance is comparable to the state-of-the-art NWM, WW3-ST6. This further validates the effectiveness of our AI model. In contrast, for the models that used assimilation, there was a substantial improvement in all error metrics, demonstrating the effectiveness of assimilation in AI modeling. The following four figures have now been included in the revised manuscript and Supporting Information.

[Figure]

*Figure R7.* *The comparison between SWHs from the AI model and NDBC Buoy in 2020 for global ocean. (a) The scatter plot between the SWHs from the two datasets. (b-e) The spatial distributions of CC, bias, RMSE, and SI, respectively.*

[Figure]

***Figure R8.*** *The same as Figure R7, but the AI model has assimilated the data from CCI-Sea State every six hours.*

[Figure]

***Figure R9.*** *The same as Figure R7, but the comparison is between the WW3-ST6 and NDBC Buoy.*

[Figure]

**Figure R10.** *The same as Figure R7, but the comparison is between the ERA5 and NDBC Buoy.*

*Discussion:*

*It will be helpful if the authors can be more specific about the suitable applications with the AI SWH model. (e.g., time scales of the operational wave forecast, locations, seasons etc.)*

We have put the discussions of the potential applications of the AI model in Section 5 (Concluding Remarks). Which reads: "An important advantage of the AI SWH model proposed here is its low computational cost compared to traditional NWMs. For example, on a personal laptop equipped with a single RTX 3060 GPU, the AI model can perform a 1-year global SWH rolling simulation at a resolution of 0.5° × 0.5° × 1h in just 10 minutes. In contrast, traditional NWMs, such as the WAVEWATCH III model, typically require several days to complete a simulation with the same output, even on supercomputing facilities. This makes the AI model particularly valuable in time-sensitive and resource-constrained scenarios, where it can be used as a surrogate for the NWMs. One potential application of this model is ensemble modeling,

both in operational wave forecasting and wave climate studies. In these applications, it is challenging to run NWMs multiple times using wind fields from different ensemble members of weather forecast models (for wave forecasting) or of various climate scenarios for long-term projection (for wave climate projection) due to the limitation of computational resources. In contrast, these tasks can be efficiently completed using the AI model, even on a standard laptop." These can be regarded as the potential applications of all AI wave models.

Besides, in the last paragraph, we wrote that: "We have demonstrated that the current SWH field and the wind field at the next time step are minimum requirements for the inputs of an AI SWH model. Such simplicity of model inputs and outputs makes this model a potential baseline for AI-based modeling of global SWH." This can be regarded as a special application for this specific model.

Regarding the time scales of the operational wave forecast, it is dependent on the performance of forecasting wind fields, while the ability of weather forecast is clearly beyond the scope of this study. Regarding the locations and seasons, as discussed in the manuscript, this AI model is more suited for wind-sea dominated conditions. Therefore, if a region has active wind fields (in some seasons), it will be suited for the application of the AI model (in the season that wind fields are active).

---

## Author Response (AR3)

**Response Letter**

Dear Editor,

Thank you for handling our manuscript submitted to GMD. We sincerely appreciate the time and effort that you and the reviewers have dedicated to evaluating our work. The reviewers' comments were highly constructive and have helped us improve the quality and clarity of the manuscript. In response, we have carefully revised the paper in accordance with their suggestions.

We hope that this version of the manuscript is acceptable for publication in GMD.

If you have any questions, please feel free to contact us.

We look forward to your positive response.

Sincerely,
Haoyu Jiang, Ph.D.
College of Life Science and Oceanography, Shenzhen University
Email: Haoyujiang@szu.edu.cn

**Response to Reviewer 3:**

*For the record, this is the second version of the manuscript I have reviewed, I did not review the original version*

*I appreciate the effort of the authors to review revise the manuscript, and the structural and positive way they addressed my concerns. The remaining differences are more rooted in the fact that we are bringing two fields together here. Those of classical physical wave modeling and those of classical data driven modeling. This will lead to different angles of the problem we look at and sometimes different conclusions. In the bigger view of this, this is a discussion that will take time as the fields get to work better together. This paper should be seen as a leap forward in data driven wave modeling and should not be held back because as communities, we are learning each other's expertise and idiom. In that context these availability of our discussion in the GMD publication process should add to the core value of the publication by itself. With this I gladly recommend the present manuscripts for publication.*

*I have one minor suggestion to correct a correction I requested in the previous review and will expand on two of my previous comment considering the discussion provided by the authors.*

We sincerely thank the reviewer for the careful reading of our revised manuscript and for the constructive and thoughtful feedback. We are very grateful for the recognition of our efforts in addressing previous concerns, and for the recommendation to accept the manuscript for publication. We especially appreciate the reviewer's insightful remarks on the broader context of classical physical modeling and data-driven approaches

*1) WW3 references: I appreciate that the manual reference is replace by Tolman (1989). However, the latter is an internal report rather than a peer reviewed paper. A better foundational reference for the WAVEWATCH series of models would be the first peer-reviewed publication (Tolman 1991, JPO 21, 782-797). With WW3 using different governing equations a separate foundational reference would be appropriate, for which Tolman et al. 2002, Weather and Forecasting, 17, 311-333 is usually used.*

We have revised the manuscript to replace the reference to the WW3 manual and the internal report (Tolman, 1989) with more appropriate peer-reviewed publications (Tolman, 1991; Tolman et al., 2002) We appreciate the reviewer's suggestion in helping us improve the rigor and completeness of our citations.

*2) The nature of the ERA5 dataset. It is well-known that we have insufficient wave observations to create a reanalysis that is data-dominated. This implies that the ERA5 data is more accurate at locations where data is or has recently been available, and less accurate at locations where no recent data has been available. This results in an anisotropic error structure. The same can be expected to be the case for the data driven model presented here. If the data that has been assimilated in ERA5 is used to validate the data driven model you effectively validate the model where it inherently is most accurate. The resulting error will then not be representative for the entire domain. If, in contrast, the data driven model was to emulate a pure hindcast, it will likely show a larger error against independent data. That error, however, will be representative for the entire model. Whereas this subtlety may become important when the data driven models become mature, it is not a major issue for this initial (successful) attempt to create such a data driven model.*

Regarding the nature of the ERA5 dataset, we are grateful for the reviewer's clarification of the error structure in reanalysis data and the implications it has for model validation. On this point, we may hold a slightly different view (though not necessarily the definitive one), which we offer for your consideration. The core principle of an AI model lies in learning statistical associations between inputs and outputs, rather than simply "memorizing" the input-output pairs. The non-uniform characteristics introduced by data assimilation—stemming from the randomness of both assimilation locations and observational errors—can be regarded as a form of random error. Consequently, training an AI model on data that contains randomly anisotropic errors does not inherently lead the model to learn or reproduce that anisotropy. In other words, a data-driven model will not inherit the anisotropic error structure present in datasets such as ERA5. Indeed, in our independent test set, the model has no means of knowing where data assimilation has occurred in ERA5, as it has never "seen" any ERA5 data from the test set.

Therefore, although ERA5 may exhibit spatiotemporal heterogeneity in error due to the sparse distribution of assimilated observations, such heterogeneity is unlikely to negatively affect the training of our AI model. On the contrary, data assimilation generally improves the overall accuracy of the model outputs, which is precisely why we chose ERA5 as the training dataset. In our previous revision, we also validated the AI model using independent buoy observations, which confirmed the overall good performance of of AI model.

*3) My note on variance errors and a Taylor diagram. I agree that the variance error is defined by the error metrics used here. I mention it because representation of minima and maxima is a critical metric when a model is used by forecasters. The latter is closely related to the variance error of the model, and hence ahs value to be presented separately.*

We also appreciate the reviewer's expansion on the importance of variance errors and the role of metrics such as those visualized in a Taylor diagram. We agree that accurately representing variability, especially extremes, is essential when models are applied in forecasting contexts. As mentioned in our previous response, while the Taylor diagram is indeed a valuable tool for model evaluation, scatter plots may provide a more detailed assessment when evaluating a single model. In particular, scatter plots can offer clearer insights into the errors associated with both minima and maxima, which are not explicitly captured by the Taylor diagram.

Once again, We sincerely thank the reviewer for the constructive suggestions and the positive recommendation on our revised manuscript.

**Response to Reviewer 4:**

*I thank the authors for their detailed response and explanation. I mostly have minor editorial comments, which are made directly in the PDF of the manuscript (attached after this review report). Here, I summarize my main editorial comments (1-3) and list a final suggestion (4) regarding one easy analysis to strengthen the paper further. Once these comments are addressed, I think the paper will be ready for acceptance.*

We sincerely thank the reviewer for taking the time to carefully read our revised manuscript and for providing such detailed and constructive feedback. We greatly appreciate your thoughtful comments, which have been extremely helpful in improving the quality of the paper.

A point-by-point response to the review comments is provided below. A substantial number of the suggestions made in the annotated PDF have also been incorporated into our revision. Please refer directly to the updated manuscript and the tracked-changes file for details.

We hope that our revisions have addressed all remaining concerns and that the manuscript is now suitable for acceptance.

*1.Formatting:*
*-Citations in the Introduction: there are two types of format issues when citing references. a) LastName et al., YYYY should be used, but sometimes the comma after et al. is missed (e.g., on line 91, Ardhuin et al. 2019 ; on line 115, Hersbach et al. 2020 ). b) sometimes, "et al" is missed for references with multiple authors. For example, on line 144 and line 149, Liu (2021) is used, but should be Liu et al. (2021). I don't have time to mark out all the instances, but the authors need to check and fix thoroughly throughout the manuscript. It should be straightforward if the authors are using a citation tool.*

*-Change U10 and V10 to $U_{10}$ and $V_{10}$ ,which are more commonly seen in the literature.*

We thank the reviewer for pointing out the formatting issues in the citations and variable notation. We have carefully gone through the entire manuscript and corrected all instances of improper citation formatting. We have also updated the notation of U10 and V10 to $U_{10}$ and $V_{10}$ in the manuscript.

*2. Organize information presented in the Results section better.*
*- Breaking the Result section into subsections to help readers follow better: I added some suggestions directly on the PDF.*
*- For Fig. 2, I think it is easier to describe results with and without data assimilation in two consecutive paragraphs, without jumping back and forth between Fig. 2 and Fig. 3. So, I suggested moving the paragraph starting on line295 to after line 281.*

Thank you for your advice, we have reorganized the content by breaking the results section into subsections to improve readability and flow. We have also followed your suggestion on the paragraph order: the discussion related to Fig. 2 with and without data assimilation has been restructured into two consecutive paragraphs. These changes help make the manuscript clearer, and we greatly appreciate your detailed feedback.

*3. Reflect the response to some of my previous questions/comments in a concise sentence in the manuscript wherever the authors think applicable:*
*I appreciate that the authors did some extra testing to address my comments. While I agree that they do not need to include the figures attached in the response, it would be nice to add a concise sentence in the related section to show and assure readers that they have done robust testing. For example:*

*- Regarding my question about choosing 2020 as the test year, I suggest the authors adding "Note that evaluating against other untrained years yields similar results, with differences in correlation coefficient (CC) and root mean square error (RMSE) being less than 0.003 and 0.03, respectively.", perhaps at the end of Line 268.*
*- For other questions, I leave it to the authors to decide whether adding extra explanation to readers in one sentence would help improve the understanding/credibility of the paper.*

We thank the reviewer for the thoughtful suggestion. Following your advice, we have added a concise sentence in the relevant section to indicate the robustness of our testing regarding the selection of the test year. For other related issues, we have carefully considered their relevance and clarity and, where appropriate, have provided brief explanations, as follows:

Added description in the subsection 2.2.3: "…Using our training samples (data from 2000 to 2017), training took approximately one hour per epoch on an NVIDIA RTX 4090 GPU. It is also tested that the model's performance will be slightly worse if the training samples are significantly reduced. Once trained, the model requires less than 10 minutes to compute (infer) the global SWH for one year at a spatio-temporal resolution of $0.5° \times 0.5° \times 1h$ on an NVIDIA RTX 3060 GPU."

Added description in the last paragraph of the subsection 3.2.1: "… A simple visual inspection of the movie indicates that the AI model effectively captures SWH evolution, suggesting that the AI model could serve as an effective surrogate for NWMs, at least for some wind-sea-dominated regions. The spatial distribution of error metrics varies in different seasons, and such variability shows the same pattern in AI-based models and NWMs."

*4. Regarding the attribution of the larger CC errors in the AI-model to swell pools: I think the authors can support this interpretation further (beyond citing references) by showing (or*

*binning) the error metrics as a function of the misalignment angle between wind and dominant or mean waves. Following the authors' argument, we could expect that on average, the CC error metric reduce with this misalignment angle between wind and waves (absolute value from 0° to 180°). Note that 0-45 is aligned/wind seas, 45-135 is cross-seas, and >135 is swell opposing wind at least commonly used in the hurricane-wave community (Holthuijsen et al. 2012; https://doi.org/10.1029/2012JC007983). I think this should be easy to do, since the mean wave direction can be obtained from ERA5 (and likely also from CCI-Sea State dataset).*

*I think adding this information in a figure will strengthen this paper. In fact, one concern I still have with the current version of the manuscript is that the results section is populated by a bunch of figures in the same manner simply for different data sources or with/without data assimilation (Fig. 4-9). (It is a bit excessive such that it seems Fig 7 and 8 can also be supplementary information.) If the key message of Fig 7 and Fig 8 is just that data assimilation reduces the overall errors, especially in the swell-dominated regions. It could probably also be summarized by the extra figure I suggested above (i.e., plotting the error metrics as a function of the misalignment angle between wind and dominant or mean waves). The authors would also need to keep in mind my comment #2 above when adding this figure. The order of information in the Results section may need to be slightly re-arranged if this extra figure is included.*

We appreciate the reviewer's insightful suggestion regarding the attribution of CC errors to swell-dominated conditions. Following your recommendation, we added an analysis showing the variation of correlation coefficient (CC) as a function of the swell energy proportion, which we believe provides a more direct and reliable indication of swell influence than the misalignment angle between wind and wave directions due to several reasons:

1) In many ocean regions, multiple wave systems (or partitions) often coexist. In such cases, neither the mean wave direction nor the peak wave direction can fully represent the overall wave field, and relying solely on these metrics may even be misleading.

2) Even when wave energy is concentrated in a single direction, the alignment between wind and wave directions does not necessarily indicate that the sea state is wind-sea dominated. The classification of wind sea versus swell typically requires an analysis of the relationship between wind speed (sometimes, its component projected onto the wave direction) and the phase speed of the waves.

3)When wind and wave directions are misaligned, a directional difference of 180° is not inherently more indicative of swell conditions than a difference of 90°.

ERA5 provides separate records of wind sea and swell significant wave heights based on spectral partition technology, allowing us to compute the swell energy proportion as $SWH_{swell}^2/SWH_{total}^2$. We have added a new figure to illustrate how the model performance (in terms of CC) changes with this ratio. The new figure and corresponding explanation have been added to the corresponding paragraphs, the revised manuscript is described as follows:

[Figure]

**Figure 5.** Correlation coefficient between AI model and ERA5 data as a function of swell energy proportion the global ocean in the year of 2020. The orange and blue lines represent the AI model before and after data assimilation, respectively, and the grey bars indicate the variation in sample size as a function of swell energy proportion.

"…However, in the tropical oceans, especially along their eastern coasts where swells are predominant ("swell pools", Chen et al., 2002), the CCs are below 0.9 (~0.85 in the Indian Ocean, ~0.8 in the Atlantic Ocean, and ~0.7 in the Pacific Ocean). To further examine whether these results are related to the presence of swell, we examined the relationship between the swell energy proportion (i.e., the ratio of the square of the swell SWH to the square of the total SWH) and the CC across the global ocean (the orange line in Fig. 5). The results show a clear trend: the smaller the swell energy proportion, the higher the CC. In particular, when the proportion is below 0.7, the CC values are consistently above 0.99, indicating robust model performance in wind-sea-dominated regions. However, when the swell energy proportion

exceeds 0.7, the CC values for the model without data assimilation drop significantly, corroborating its lower performance in swell-dominated regions."

And added description in the last paragraph of the subsection 3.2.2: "…the overall errors become significantly smaller, particularly in the swell-dominated regions, with assimilation. These results are further supported by Fig. 5, where the model with data assimilation consistently maintains substantially higher CC values, even when the swell energy proportion exceeds 0.7. Similar to Supplementary Movie S1, the comparison animation of the results with assimilation is placed in Supplementary Movie S2, …"

And as per your recommendation, we have transferred the original Figs. 7 and 8 to the Supporting Information, where they are now designated as Figs. S7 and S8.

**This comment is from annotated PDF (Line 285):** *the same should also be true for the hot-start experiment, right? So it is interesting that this variability in the error metrics is also larger in the cold-start experiment before errors stabilize (blue shading being wider than the orange).*

Yes. As you point out, the variance of the error metrics is indeed wider in cold-start experiments, especially in the early stages of prediction before the error stabilizes. This wider spread (blue shading) reflects the fact that cold-start methods are more sensitive to the initial wind field due to the lack of a priori information. In contrast, hot-start experiments benefit from the availability of a wave initial field, which can produce more stable results.

---

## Author Response (AR4)

**Response Letter**

Dear Editor,

We are very grateful for your handling of our manuscript submitted to GMD. We sincerely appreciate the time and effort you and the reviewers have devoted to evaluating our work throughout the review process.

We are pleased to learn that the revised version has been deemed suitable for publication subject to technical corrections. As suggested, we have included an acknowledgment to all anonymous reviewers in the Acknowledgments section of the manuscript.

Thank you again for your support and guidance during the review.

We look forward to the final publication of our work in GMD.

Sincerely,
Haoyu Jiang, Ph.D.
College of Life Science and Oceanography, Shenzhen University
Email: Haoyujiang@szu.edu.cn